# Knockdown of PHOX2B in the retrotrapezoid nucleus reduces the central $CO_2$ chemoreflex in rats

Silvia Cardani[1,2†], Tara A Janes[1,2†], William Betzner[1‡], Silvia Pagliardini[1,2,3]*

[1]Department of Physiology, Faculty of Medicine and Dentistry, University of Alberta, Edmonton, Canada; [2]Women and Children's Health Research Institute, University of Alberta, Edmonton, Canada; [3]Neuroscience and Mental Health Institute, University of Alberta, Edmonton, Canada

*For correspondence:
silviap@ualberta.ca

[†]These authors contributed equally to this work

Present address: [‡]Departments of Clinical Neurosciences and Community Health Sciences, Hotchkiss Brain Institute, University of Calgary Cumming School of Medicine, Calgary, Canada

Competing interest: The authors declare that no competing interests exist.

**Abstract** PHOX2B is a transcription factor essential for the development of different classes of neurons in the central and peripheral nervous system. Heterozygous mutations in the PHOX2B coding region are responsible for the occurrence of Congenital Central Hypoventilation Syndrome (CCHS), a rare neurological disorder characterised by inadequate chemosensitivity and life-threatening sleep-related hypoventilation. Animal studies suggest that chemoreflex defects are caused in part by the improper development or function of PHOX2B expressing neurons in the retrotrapezoid nucleus (RTN), a central hub for $CO_2$ chemosensitivity. Although the function of PHOX2B in rodents during development is well established, its role in the adult respiratory network remains unknown. In this study, we investigated whether reduction in PHOX2B expression in chemosensitive neuromedin-B (NMB) expressing neurons in the RTN altered respiratory function. Four weeks following local RTN injection of a lentiviral vector expressing the short hairpin RNA (shRNA) targeting *Phox2b* mRNA, a reduction of PHOX2B expression was observed in *Nmb* neurons compared to both naive rats and rats injected with the non-target shRNA. PHOX2B knockdown did not affect breathing in room air or under hypoxia, but ventilation was significantly impaired during hypercapnia. PHOX2B knockdown did not alter *Nmb* expression but it was associated with reduced expression of both *Task2* and *Gpr4*, two $CO_2$/pH sensors in the RTN. We conclude that PHOX2B in the adult brain has an important role in $CO_2$ chemoreception and reduced PHOX2B expression in CCHS beyond the developmental period may contribute to the impaired central chemoreflex function.

## eLife assessment

This **important** study utilizes a viral-mediated short hairpin RNA (shRNA) approach to investigate in a novel way the role of the wild-type PHOX2B transcription factor expressed in critical chemosensory neurons in the brainstem retrotrapezoid nucleus (RTN) region for maintaining normal CO2 chemoreflex control of breathing in adult rats. The **convincing** results show blunted ventilation during elevated inhaled CO2 (hypercapnia) with knockdown of PHOX2B, accompanied by a reduced expression of Gpr4 and Task2 mRNA for the proposed RTN neuron proton sensor proteins GPR4 and TASK2. These results indicate that maintained expression of wild-type PHOX2B affects respiratory control in adult animals, complementing previous studies showing that PHOX2B-expressing RTN neurons may be critical for chemosensory control throughout the lifespan, and with implications for neurological disorders involving the RTN, which will be of interest to neuroscientists studying respiratory neurobiology as well as the neurodevelopmental control of motor behavior.

## Introduction

The Paired Like Homeobox 2B (PHOX2B) is a transcription factor essential for embryonic differentiation and for the maintenance of the neuronal phenotype of different classes of neurons in both the central and peripheral nervous systems (*Brunet and Pattyn, 2002*; *Pattyn et al., 1997*; *Pattyn et al., 1999*; *Pattyn et al., 2000*; *Stanke et al., 1999*; *Tiveron et al., 1996*). In recent years, the PHOX2B gene has been extensively investigated, not only for its neurodevelopmental role, but also because its heterozygous mutation leads to Congenital Central Hypoventilation Syndrome (CCHS, OMIM 209880; *Amiel et al., 2003*), Hirschsprung's disease (HSCR; OMIM 142623; *Berry-Kravis et al., 2006*; *Trochet et al., 2005*), and neuroblastoma (*Bourdeaut et al., 2005*; *van Limpt et al., 2004*). CCHS is a rare (1:200,000 births in France *Trang et al., 2005*; 1:148,000 births in Japan *Shimokaze et al., 2015*) disorder characterised by a failure to respond to both hypercapnia and hypoxia (*Amiel et al., 2003*; *Di Lascio et al., 2020*; *Weese-Mayer et al., 2017*). The most frequent PHOX2B mutation is an expansion of a polyalanine repeat, ranging from +5 to +13 residues on the C-terminal of PHOX2B, an area that regulates DNA binding affinity and protein solubility (*Amiel et al., 2003*; *Di Lascio et al., 2016*; *Weese-Mayer et al., 2003*). CCHS symptoms present with varying degrees of severity depending on the expansion of the mutation (*Berry-Kravis et al., 2006*; *Di Lascio et al., 2018b*; *Matera et al., 2004*). In most mild cases, patients display normal ventilation during wakefulness, hypoventilation during sleep leading to hypercarbia and hypoxemia, as well as a lack of sensitivity and discomfort to these challenges. In the most severe of cases, hypoventilation may also be present while awake.

Respiratory disturbances resulting from the PHOX2B mutations appear to stem from its widespread expression in respiratory-related neurons both during development and into adulthood. Although PHOX2B is absent in respiratory rhythmogenic neurons of the preBötzinger complex, it is highly expressed in adult neurons that are responsible for $CO_2$ and $O_2$ chemoreflexes (*Kang et al., 2007*). Among those are the neurons of the retrotrapezoid nucleus (RTN), the locus coeruleus, the carotid bodies and the nucleus of the solitary tract that, together with the dorsal motor nucleus of the vagus nerve and the area postrema, make up to the dorsal vagal complex, a structure that provides the major integrative centre for the mammalian autonomic nervous system (*Cutsforth-Gregory and Benarroch, 2017*; *Guyenet et al., 2019*; *Kang et al., 2007*; *Zoccal et al., 2014*). Previous work by our group and others specifically investigated the role of PHOX2B-expressing RTN neurons by either selective lesions or inactivation of neuronal activity of these neurons (*Abbott et al., 2011*; *Basting et al., 2015*; *Marina et al., 2010*; *Nattie and Li, 1994*; *Souza et al., 2018*; *Souza et al., 2023*; *Janes et al., 2024*) and showed that RTN neurons have a key role in central $CO_2$ chemoreception.

Defects observed in the $CO_2$-chemoreflex of CCHS transgenic rodent models are primarily caused by the improper development or function of PHOX2B-expressing neurons in the RTN (*Dubreuil et al., 2008*; *Ramanantsoa et al., 2011*), although anatomo-pathological studies in CCHS patients suggest that the respiratory impairment may be caused by a more widespread brain dysfunction (*Harper et al., 2015*; *Nobuta et al., 2015*). Transgenic mice born with PHOX2B knockout fail to develop the RTN, the locus coeruleus (*Pattyn et al., 1999*) and catecholaminergic groups in A1, A2, A5, and A7 (*Pattyn et al., 2000*), as well as neurons of the nucleus of the solitary tract and the carotid bodies. Mice harbouring heterozygous PHOX2B knockout develop apparently normal but have depressed responses to hypoxia and hypercapnia (*Dauger et al., 2003*) and sleep disordered breathing in the perinatal period (*Durand et al., 2005*). Furthermore, the expression of the most common heterozygous PHOX2B mutation found in CCHS patients (PHOX2B+7 Ala expansion) in mice results in a near complete loss of RTN neurons, impaired $CO_2$ response and consequent death in the newborn period (*Dubreuil et al., 2008*; *Madani et al., 2021*; *Ramanantsoa et al., 2011*; *Takakura et al., 2008*). More selective expression of the mutant PHOX2B in the RTN neurons allowed for survival in the neonatal period and partial recovery of the $CO_2$ response in adulthood (*Ramanantsoa et al., 2011*).

Multiple potential cellular mechanisms have been implicated in the aetiology of CCHS, from loss of PHOX2B function in development due to heterozygous mutant PHOX2B expression, to gain of function of the mutated protein that may alter transcriptional activity in neurons, as well as aggregate formation and premature cell death (*Bachetti et al., 2005*; *Di Lascio et al., 2013*; *Dubreuil et al., 2008*; *Durand et al., 2005*; *Goridis et al., 2010*; *Trochet et al., 2005*). While its role during embryonic neurodevelopment is established, PHOX2B is still expressed in selected neuronal populations in the adult brain (*Kang et al., 2007*; *Stornetta et al., 2006*) and its function, with the exception of the contribution to the maintenance of the noradrenergic phenotype (*Fan et al., 2011*), is not yet fully

understood. Considering the fundamental role of PHOX2B in CCHS pathogenesis and its expression in key structures for $CO_2$ chemoreflex responses, it is essential to understand how both wild-type and mutant PHOX2B proteins impact $CO_2$ homeostasis and ventilation not only across development, but also in adulthood.

In order to have a better understanding of the role of PHOX2B in the $CO_2$ homeostatic processes, we used a non-replicating lentivirus vector of two short-hairpin RNA (shRNA) clones targeting selectively *Phox2b* mRNA to knockdown the expression of PHOX2B in the RTN of adult rats and tested ventilation and chemoreflex responses. In parallel, we also determined whether knockdown of PHOX2B in adult *Nmb* neurons of the RTN negatively affected their cell survival. Finally, we sought to provide a mechanistic link between PHOX2B expression and the chemosensitive properties of *Nmb* neurons of the RTN, which have been attributed to two proton sensors, the proton-activated G-protein-coupled receptor (GPR4) and the proton-modulated potassium channel (TASK-2; *Gestreau et al., 2010*; *Guyenet et al., 2016*; *Kumar et al., 2015*). The expression of these sensors is partially overlapping in *Nmb* neurons of the RTN and the genetic deletion of either one impairs the central respiratory chemoreflex with maximal impairment when both sensors are eliminated (*Gestreau et al., 2010*; *Guyenet et al., 2016*; *Kumar et al., 2015*).

Our results indicate that progressive knockdown of PHOX2B in RTN causes a reduction in the chemoreflex response that was proportional to the decrease in the fraction of *Nmb*/PHOX2B expressing RTN neurons. This effect was associated with a reduction in the expression of *Gpr4* and *Task2* mRNA in *Nmb* expressing cells of the RTN in which PHOX2B was knocked down, suggesting a role for PHOX2B in the transcription of the proton sensors in RTN neurons.

## Results

To directly investigate the role of PHOX2B in RTN neurons, we performed selective knockdown of this protein in adult rats through local injection of a non-replicating lentiviral vector of two short-hairpin RNA (shRNA) clones targeting two different sequences of the *Phox2b* mRNA (n=25) or non-target shRNA as control (NT-shRNA; n=23). Initial experiments utilized 200 nl/injection at each of four stereotaxic coordinates ($1 \times 10^9$ VP/ml; n=14). Four weeks post injection, we assessed ventilatory parameters in room air, hypercapnia, and hypoxia. Interestingly, a small but significant impairment in the $CO_2$ chemoreflex was observed not only in PHOX2B-shRNA but also in NT-shRNA rats compared to pre-surgical baseline (allometric $V_E$ in 7% $CO_2$: –21% and –24%, respectively; *Figure 1—figure supplement 1*). Because histological analysis also showed significant reduction of $Nmb^+$/$PHOX2B^+$ RTN cells in NT-shRNA and PHOX2B-shRNA rats (33% and 19% cells remaining, respectively; *Figure 1—figure supplement 1*; *Supplementary file 1*), we concluded that the volume and titre of the shRNA viral constructs triggered off-target effects (*McBride et al., 2008*; *van Gestel et al., 2014*) that cause cell death in RTN neurons.

### PHOX2B expression is modestly reduced two weeks following infection without respiratory impairment

Due to the off-target effects obtained with large volume injections, we reduced injection volume of both PHOX2B shRNA (n=17) and NT-shRNA viruses (n=17; $4 \times 100$ nl/injection; $1 \times 10^9$ VP/ml).

Two weeks post-surgery we assessed respiratory function during exposure to room air, hypercapnia and hypoxia. Overall, we observed little change in respiration at this time point. During room air, as well as in 5% and 7.2% $CO_2$, frequency ($f_R$), was reduced in every experimental group (naive, NT-shRNA, PHOX2B-shRNA) compared to baseline (Air: p=0.014; 5% $CO_2$: p<0.001; 7.2% $CO_2$ p<0.001; *Figure 1A*; *Supplementary file 1*) but no effect on treatment was observed, suggesting that these changes were likely due to the increase in age and body weight during the long-term experiment (naive:+32.2%; NT-shRNA:+27.8%; PHOX2B-shRNA:+28.6% change in body weight). Tidal volume was reduced in the PHOX2B-shRNA rats at 7.2% $CO_2$ compared to both naive rats (naive: $9.6 \pm 1.2$ ml·kg$^{-1}$; PHOX2B-shRNA: $8.5 \pm 0.9$ ml·kg$^{-1}$; –11%; p=0.022) and baseline pre-surgery conditions (baseline: $9.6 \pm 1.3$ ml·kg$^{-1}$; –11%; p<0.001; *Figure 1B*) but it was not different from NT-shRNA.

Despite similar changes in body weight across treatment groups, allometric ventilation ($V_E$ ALLO) was higher in naive rats compared to the other treatment groups in room air (naive: $38.4 \pm 4.5$ ml·min$^{-1}$; NT-shRNA: $34.7 \pm 2.8$ ml·min$^{-1}$; PHOX2B-shRNA: $34.0 \pm 4.5$ ml·min$^{-1}$; p=0.005; *Figure 1C*), but no changes

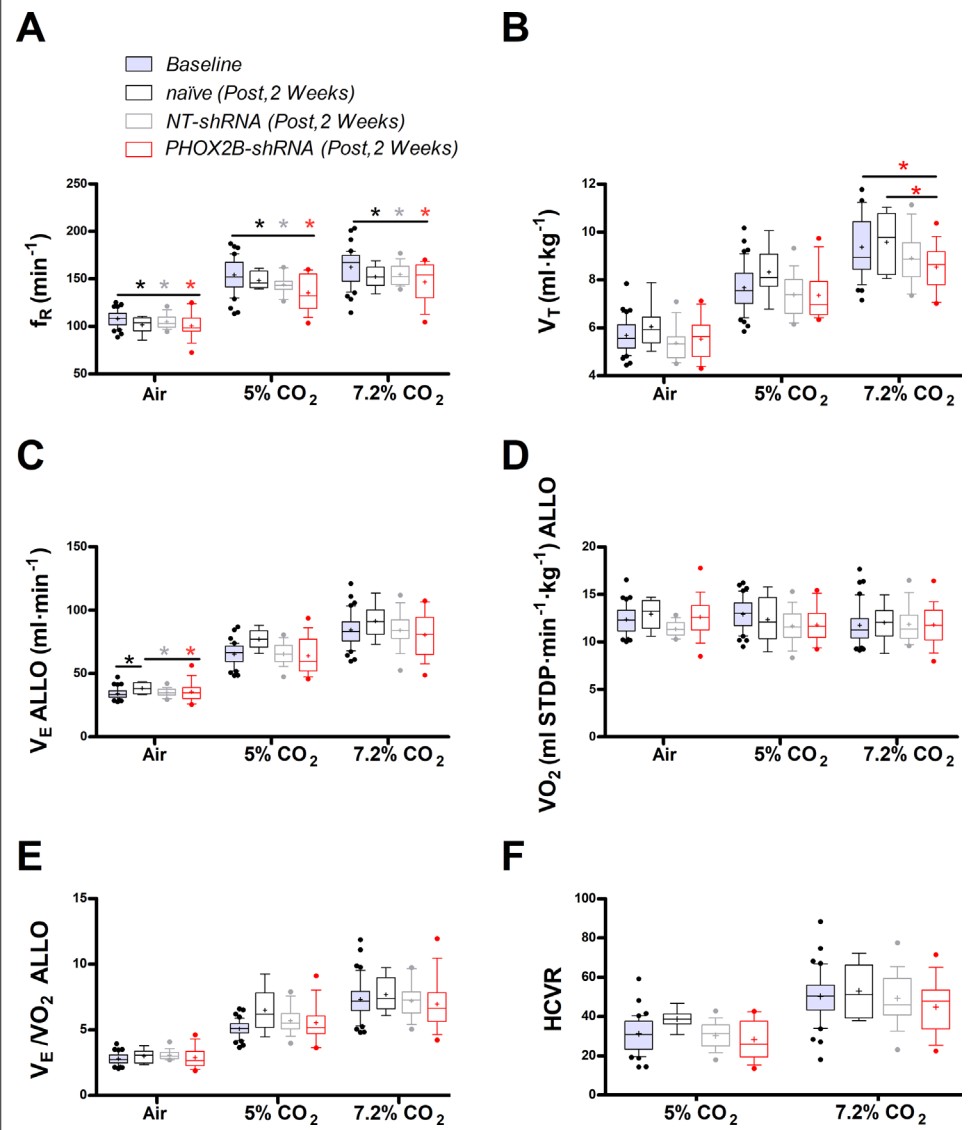

**Figure 1.** Respiratory data 2 weeks post viral PHOX2B shRNA injection. (**A**) Breathing frequency ($f_R$,), (**B**) tidal volume ($V_T$), (**C**) allometric minute ventilation ($V_E$ ALLO), (**D**) oxygen consumption ($VO_2$ ALLO), (**E**) convective requirement ratio ($V_E/VO_2$ ALLO), (**F**) hypercapnic ventilatory response (HCVR absolute change in $V_E$ ALLO vs. corresponding room air) of baseline (pre-surgery, grey filled box), naive (black n=8), non-target control shRNA (NT-shRNA, grey n=17) and PHOX2B shRNA (PHOX2B-shRNA, red n=17) rats 2 weeks post injection during room air, hypercapnia 5% and 7.2% $CO_2$. $f_R$ was equally reduced in all experimental groups compared to baseline but no treatment effect was observed (**A**). $V_T$ was significantly impaired following RTN injection in PHOX2B-shRNA group compared to naive animals (p=0.022) and baseline (p<0.001) at 7.2% $CO_2$ (**B**). $V_E$ ALLO was increased in naive rats compared to the other treatment groups (p=0.005) in room air (**C**). Boxplots: median, 1st – 3rd quartiles and 10th – 90th percentiles, outliers = dots, '+' indicates arithmetic mean. Bonferroni post-hoc as indicated. Black*, different from naive; Grey*, different from NT-shRNA; Red*, different from PHOX2B- shRNA.

The online version of this article includes the following source data and figure supplement(s) for figure 1:

**Source data 1.** Respiratory data 2 weeks post viral PHOX2B shRNA injection.

**Figure supplement 1.** Respiratory and anatomical data 4 weeks post large viral PHOX2B shRNA injection.

in $V_E$ ALLO were observed in hypercapnia (7.2% $CO_2$) across treatments (naive: 91.5±13.2 ml·min$^{-1}$; NT-shRNA: 84.4±13.7 ml·min$^{-1}$; PHOX2B-shRNA: 80.5±17.2 ml·min$^{-1}$). Oxygen consumption ($VO_2$ ALLO) was not different between experimental conditions or treatment groups (*Figure 1D*), and $V_E/VO_2$ ALLO indicated that ventilation was matched with metabolism for each treatment across all

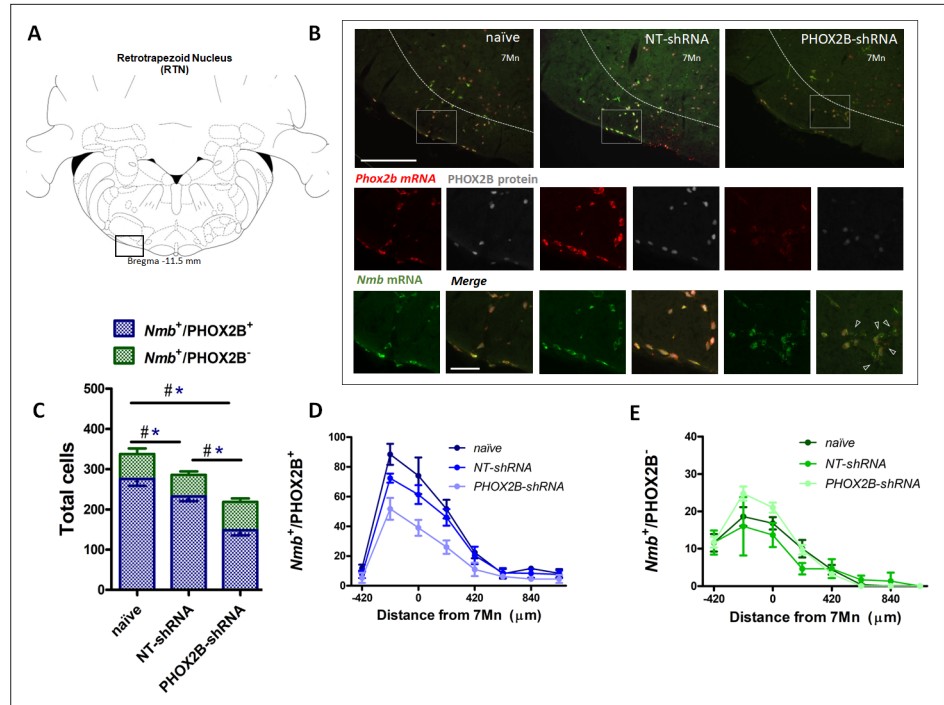

**Figure 2.** PHOX2B and *Nmb* expression and total cell count within the RTN area in naive, NT-shRNA and PHOX2B-shRNA injected rats two weeks post viral shRNA injection. (**A**) Schematic and representative image of a transverse brainstem section at the level of the RTN (–11.5 mm distance from Bregma) showing the area of investigation containing RTN neurons. (**B**) Expression of *Phox2b* mRNA (red), PHOX2B protein (white) and *Nmb* mRNA (green) in RTN *Nmb*+/PHOX2B+ and *Nmb*+/PHOX2B- neurons in naive (left), NT-shRNA (middle), and PHOX2B-shRNA (right) rats (magnified view inserts below). Arrowheads indicate absence of PHOX2B protein. Scale bar = 400 µm (top figures), 150 µm (inserts below). (**C**) The number of total cells (*Nmb*+/PHOX2B+ plus *Nmb*+/PHOX2B-) comprising the RTN were reduced in PHOX2B-shRNA rats (n=4) as compared to naive (n=4) and NT-shRNA (n=4), and in NT-shRNA rats as compared to naive rats (black#, One-way ANOVA, p<0.001). The number of *Nmb*+/PHOX2B+ cells were reduced in PHOX2B-shRNA rats as compared to naive and NT-shRNA (blue*, One-way ANOVA, p<0 0.001). The number of *Nmb*+/PHOX2B- cells was unchanged across groups. (**D,E**) Rostral-caudal distribution (distance from the caudal tip of the facial nucleus, 7Mn) of *Nmb*+/PHOX2B+ (**D**) and *Nmb*+/PHOX2B- (**E**) neurons along the RTN.

The online version of this article includes the following source data for figure 2:

**Source data 1.** PHOX2B and Nmb expression and total cell count within the RTN area in naive, NT-shRNA and PHOX2B-shRNA injected rats two weeks post viral shRNA injection.

---

gas compositions (naive: 7.7±1.3; NT-shRNA: 7.2±1.3; PHOX2B-shRNA: 6.9±2.0; *Figure 1E*). These results suggest that despite a reduction in hypercapnic $V_T$ in the PHOX2B-shRNA rats, overall ventilation was not impaired. Since we observed differences in room air $V_E$ ALLO between treatment groups, we normalized these data to show the hypercapnic ventilatory response (HCVR, i.e. absolute change in $V_E$ from room air; *Figure 1F*). The HCVR was also not affected by the PHOX2B shRNA treatment two weeks post-injection (naive: 53.1±13.4 ml·min$^{-1}$; NT-shRNA: 49.6±13.2 ml·min$^{-1}$; PHOX2B-shRNA: 46.5±14.8 ml·min$^{-1}$).

To determine the extent of PHOX2B knockdown in RTN neurons, we combined RNAScope and immunohistochemistry assays to quantify RTN neurons expressing *Nmb* and PHOX2B (*Figure 2A and B*). We reasoned that by assessing the fraction of RTN neurons that express only *Nmb* (compared to *Nmb*+/PHOX2B+), we would be able to determine the level of PHOX2B knockdown in our experiments. Two weeks post viral injection, we observed a small decrease (15.2%) in the total number of *Nmb* RTN neurons (i.e. *Nmb*+/PHOX2B+ plus *Nmb*+/ PHOX2B-) in NT-shRNA rats compared to naive rats (naive; 337.6±11.95; n=4; NT-shRNA: 286.0±10.44; n=4; p<0.001; *Figure 2C*; *Supplementary file 1*). The total number of *Nmb* cells of the RTN in PHOX2B-shRNA rats was further reduced compared to both naive rats (PHOX2B-ShRNA 218.8±7.8; n=4; –35.2%; p<0.001) and to NT-shRNA (–23.5%; p<0.001).

We then quantified the number of neurons in the RTN expressing either *Nmb* and PHOX2B (*Nmb*+/PHOX2B+) or *Nmb* only (*Nmb*+/PHOX2B-). The number of *Nmb*+/PHOX2B+ neurons in PHOX2B-shRNA rat was reduced by 36.1% compared to NT-shRNA and by 46.2% compared to naive rats (*Nmb*+/PHOX2B+ NT-shRNA: 232.3±11.6 vs naive: 275.8±17.1; vs PHOX2B-shRNA: 148.3±12.6; p<0.05 and 0.001, respectively; *Figure 2C and D*). On the contrary, the fraction of *Nmb*+/PHOX2B- neurons in PHOX2B-shRNA rats was not different compared to naive and NT-shRNA rats (*Nmb*+/PHOX2B- NT-shRNA: 53.7±8.6; vs naive: 61.8±13.9; vs PHOX2B-shRNA: 70.5±8.7 cells; *Figure 2C and E*). These results thus indicate that despite modest loss of *Nmb*+ neurons in NT-shRNA and a reduction in the proportion of *Nmb* neurons expressing PHOX2B in PHOX2B-shRNA rats, 2 weeks of PHOX2B shRNA viral infection was not sufficient to impair ventilation.

## Four weeks of PHOX2B knockdown impairs the $CO_2$-chemoreflex

Since the shRNAs integrate into the genome of the infected cells and are continuously produced, it is reasonable to expect that knockdown efficiency is time-dependent, and an extended infection time would result in a higher knockdown effect. To test this hypothesis, a subgroup of rats (naive, NT-shRNA and PHOX2B-shRNA) were investigated 4 weeks post-viral injection (*Figure 3A and B*).

The total number of *Nmb* neurons in the RTN of PHOX2B-shRNA rats (*Nmb*+/PHOX2B+ plus *Nmb*+/PHOX2B-; *Figure 3C*; *Supplementary file 1*) was significantly reduced compared to naive rats (naive: 347.8±12 cells, n=4; PHOX2B-shRNA: 249.5±33.9 cells; n=6; –28.3%; p=0.009) but no difference was observed compared to NT-shRNA rats (294.9±52.8 cells; n=10; –15.4%;). Furthermore, there was no difference in the total number of *Nmb* cells in RTN between 2 and 4 weeks post infection for both NT-shRNA (week 2: 286±10.4; week 4: 294.9±52.8, p=0.783) and PHOX2B-shRNA rats (week 2: 218.8±7.8; week 4: 249.5±33,9, p=0.118), suggesting that the progressive PHOX2B knockdown was specific for PHOX2B-shRNA treatment and it was not accompanied by an increased cell death beyond the first 2 weeks.

The number of *Nmb*+/PHOX2B+ cells in the RTN was significantly reduced in PHOX2B-shRNA rats by 67.0% and 60.7% compared to naive and NT-shRNA rats, respectively (naive: 284.5±19.8; NT-shRNA: 238.7±56; PHOX2B-shRNA: 93.8±11.9 cells, p<0.001; *Figure 3D*). Moreover, a significant increase in the fraction of *Nmb*+/PHOX2B- was observed with PHOX2B knockdown (PHOX2B-shRNA: 155.7±22.7; naive: 63.2±12 cells;+146.1% increase vs naive; NT-shRNA: 56.2±13.8 cells;+177% increase vs NT-shRNA, pm<0.001; *Figure 3E*), confirming the efficiency of the PHOX2B knockdown in the *Nmb* cells of the RTN.

The respiratory function prior to histological examination (4 weeks post-viral injection) showed significant differences in PHOX2B-shRNA rats (n=6) compared to both pre-surgery baseline, naive (n=8) and NT-shRNA rats (n=10). Similar to what we reported at 2 weeks post-surgery, $f_R$ at 4 weeks was lower in every treatment group compared to baseline recordings (Air: p=0.001; 5% $CO_2$: p<0.001; 7.2% $CO_2$ p<0.001; *Figure 4A*; *Supplementary file 1*).

During room air, $V_T$ was reduced in naive rats compared to baseline (baseline: 5.9±0.5 ml·kg$^{-1}$; week 4: 5.2±0.6 ml·kg$^{-1}$; –12%; p=0.001; *Figure 4B*). Moreover, $V_T$ in PHOX2B-shRNA rats was reduced compared to NT-shRNA (NT-shRNA: 5.6±0.4 ml·kg$^{-1}$; PHOX2B-shRNA: 5.3±0.4 ml·kg$^{-1}$; –10%; p=0.043) and to the pre-surgery baseline (6.1±0.7 ml·kg$^{-1}$; –18%; p=0.002). No significant changes were observed at 5% $CO_2$, although a decrease in $V_T$ occurred at 7.2% $CO_2$ in PHOX2B-shRNA rats compared to pre-surgery baseline (baseline: 7.7±1.2 ml·kg$^{-1}$; PHOX2B-shRNA: 6.7±0.6 ml·kg$^{-1}$; –31%; p<0.001), naive rats (7.9±0.9 ml·kg$^{-1}$; –26%; p<0.001), and NT-shRNA rats (8.1±0.7 ml·kg$^{-1}$; –23%; p=0.002).

Decreased $V_T$ observed for PHOX2B-shRNA led to a ventilation ($V_E$ ALLO) impairment during exposure to hypercapnia (7.2% $CO_2$, PHOX2B-shRNA vs. NT-shRNA: –17%; p=0.021; PHOX2B-shRNA vs. naive; –21%; p=0.007; PHOX2B-shRNA vs. baseline, –18% p=0.002; *Figure 4C*). Since $O_2$-consumption ($VO_2$ ALLO) did not change (data not shown), a significant reduction in $V_E/VO_2$ ALLO confirmed alveolar hypoventilation in PHOX2B-shRNA animals compared to NT-shRNA rats both at 5% (NT-shRNA: 5.8±0.5; PHOX2B-shRNA: 4.8±0.4; –16%; p=0.023) and 7.2% $CO_2$ (NT-shRNA: 7.9±1.2; PHOX2B-shRNA: 6.1±0.8; –24%; p=0.004; *Figure 4D*). Moreover, the HCVR (*Figure 4E*) at 7.2% $CO_2$ was lower in PHOX2B-shRNA (40.3±9.4 ml·min$^{-1}$) rats compared to the pre-surgery baseline (58.1±8.9 ml·min$^{-1}$; –31%; p=0.015), naive rats (63.6±14.3 ml·min$^{-1}$; –37%; p=0.008) and NT-shRNA rats (56.0±10.4 ml·min$^{-1}$; –28%; p=0.016). Analysis of individual rats in each treatment group over time

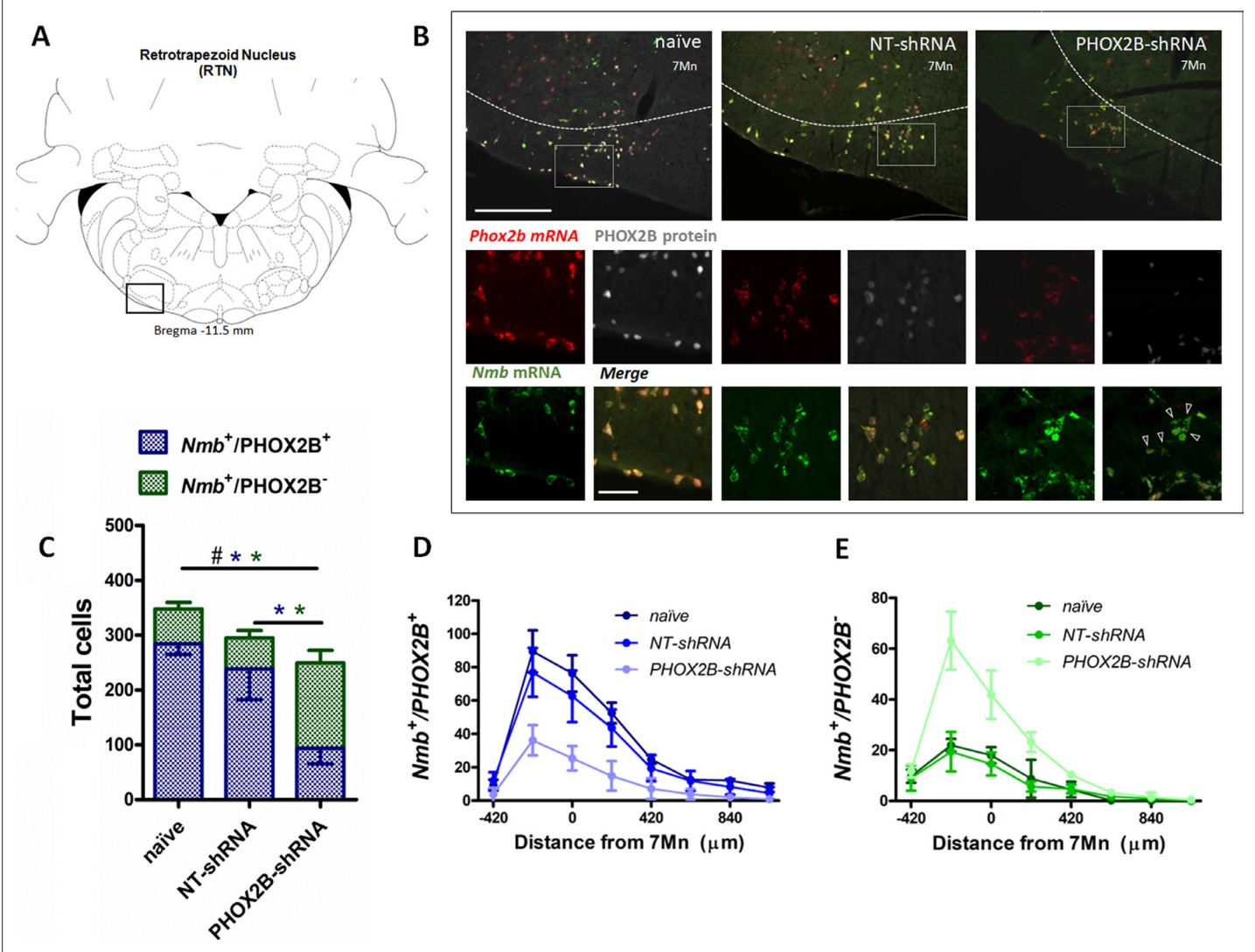

**Figure 3.** PHOX2B and *Nmb* expression and total cell count within the RTN area in naive, NT-shRNA and PHOX2B-shRNA injected rats 4 weeks post viral shRNA injection. (**A**) Schematic and representative image of a transverse brainstem section at the level of the RTN (–11.5 mm distance from Bregma) showing the area of investigation containing RTN neurons. (**B**) Expression of *Phox2b* mRNA (red), PHOX2B protein (white) and *Nmb* mRNA (green) in naive (left), NT-shRNA (middle), and PHOX2B-shRNA (right) rats (magnified view insert). Arrowheads indicate absence of PHOX2B protein. Scale bar = 400 µm (top figures), 150 µm (inserts below). (**C**) The number of total cells ($Nmb^+$/PHOX2B$^+$ + $Nmb^+$/PHOX2B$^-$) comprising the RTN were reduced in PHOX2B-shRNA rats (n=6) as compared to naive (n=4) (Black#, One-way ANOVA, p=0.0087) but not to NT-shRNA (n=10). The number of $Nmb^+$/PHOX2B$^+$ cells were reduced in PHOX2B-shRNA rats as compared to naive and NT-shRNA (Blue*, one-way ANOVA, p<0.001). The number of $Nmb^+$/PHOX2B$^-$ cells were increased in PHOX2B-shRNA rats as compared to both naive and NT-shRNA (Green*, one-way ANOVA p<0.001). (**D,E**) Rostral-caudal distribution (distance from the caudal tip of the facial nucleus, 7Mn) of $Nmb^+$/PHOX2B$^+$ (**D**) and $Nmb^+$/PHOX2B$^-$ (**E**) neurons along the RTN.

The online version of this article includes the following source data for figure 3:

**Source data 1.** PHOX2B and Nmb expression and total cell count within the RTN area in naive, NT-shRNA and PHOX2B-shRNA injected rats 4 weeks post viral shRNA injection.

showed that the HCVR was only consistently impaired in PHOX2B-shRNA rats between weeks 2 and 4 (*Figure 1F*; p=0.007).

Given that previous studies have proposed a role for the RTN in the hypoxic chemoreflex (*Barna et al., 2016*; *Basting et al., 2015*; *Gourine et al., 2005*; *Wickström et al., 2004*), and we observed impairment of the $CO_2$-chemoreflex at 4 weeks post-infection, we investigated whether PHOX2B knockdown would also affect respiratory responses to hypoxia (10% $O_2$). The 4 week $V_E$ ALLO was increased relative to baseline values in naive (baseline: 62.3±5.2 ml·min$^{-1}$; naive: 71.0±10.1 ml·min$^{-1}$;+14%),

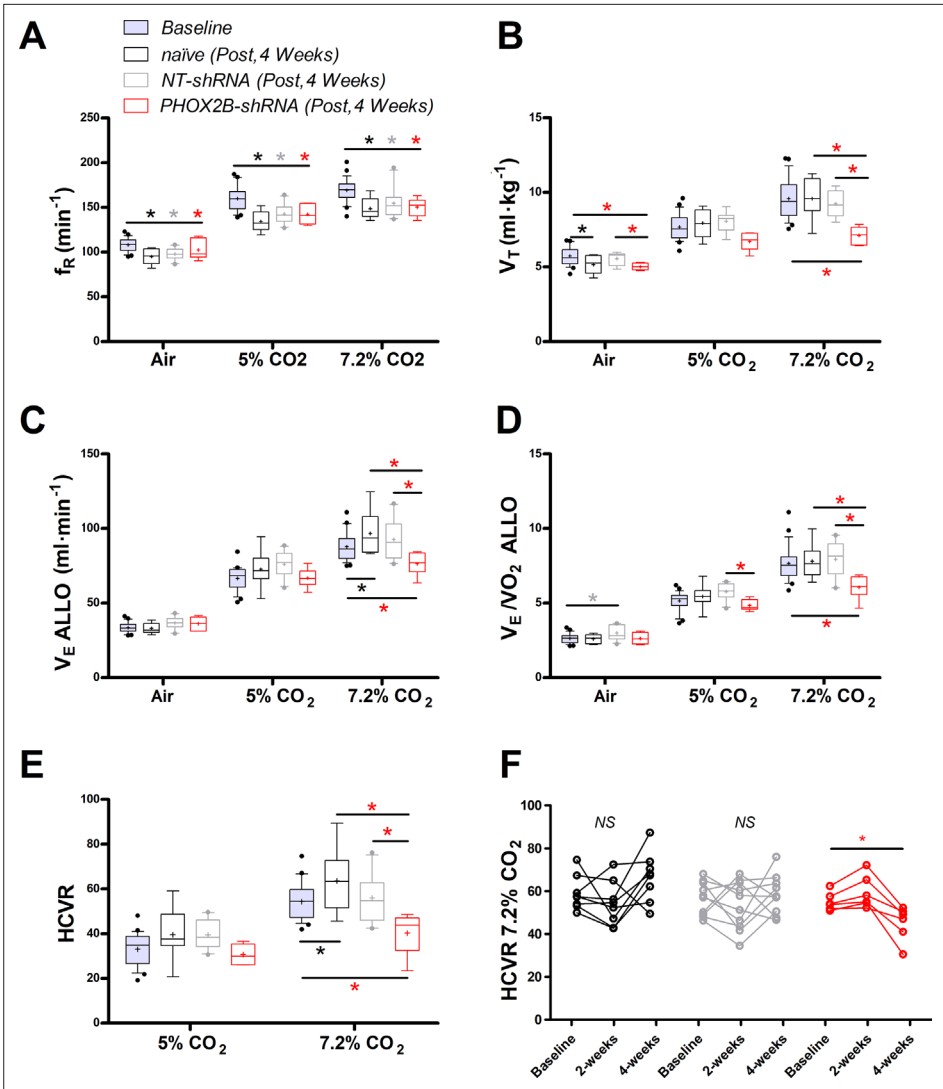

**Figure 4.** Respiratory data following 4 weeks post viral shRNA injection. (**A**) Breathing frequency ($f_R$,), (**B**) tidal volume ($V_T$), (**C**) allometric minute ventilation ($V_E$ ALLO), (**D**) convective requirement ratio ($V_E/VO_2$ ALLO), (**E**) hypercapnic ventilatory response (HCVR, absolute change in $V_E$ ALLO vs. corresponding room air), (**F**) HCVR at baseline, week 2 and week 4 post-viral injections in naive (black n=8), non-target control shRNA (NT-shRNA, grey n=10) and PHOX2B-shRNA (PHOX2B-shRNA, red n=6). $f_R$ was equally impaired in all experimental group compared to baseline but no treatment effect was observed (**A**). $V_T$ was significantly impaired following RTN injection in PHOX2B-shRNA group compared to baseline pre-surgery (p<0.001), naive rats (p<0.001), and NT-shRNA rats (p=0.002) at 7.2% $CO_2$. (**B**). $V_E$ ALLO was impaired in PHOX2B-shRNA rats during exposure to hypercapnia (7.2% $CO_2$) compared to baseline (p=0.0025), naive rats (p=0.007), and NT-shRNA rats (p=0.002). (**C**). $V_E/VO_2$ ALLO was reduced in PHOX2B-shRNA animals compared to NT-shRNA rats both at 5% (p=0.023) and 7.2% (p=0.004) $CO_2$ (**D**). HCVR during 7.2% $CO_2$ was lower in PHOX2B-shRNA rats compared to baseline (p=0.007), naive rats (p=0.001), and NT-shRNA rats (p=0.016) (**E**). Boxplots: median, 1st – 3rd quartiles and 10th – 90th percentiles, outliers = dots, '+' indicates arithmetic mean. Bonferroni post-hoc as indicated. HCVR was significantly impaired only in PHOX2B-shRNA rats 4 weeks post-surgery (One-way ANOVA p=0.007) (**F**). Black*, different from naive; Grey*, different from NT-shRNA; Red*, different from PHOX2B- shRNA.

The online version of this article includes the following source data for figure 4:

**Source data 1.** Respiratory data following 4 weeks post viral shRNA injection.

NT-shRNA (baseline: 71.8±11.1 ml·min⁻¹; NT-shRNA: 75.8±12.8 ml·min⁻¹;+6%) and PHOX2B-shRNA rats (baseline: 65.9±9.8 ml·min⁻¹; PHOX2B-shRNA: 79.3±6.4 ml·min⁻¹;+29%; p<0.001; data not shown). However, analysis of the convective requirement ratio ($V_E/VO_2$ ALLO) indicates that changes in hypoxic ventilation were due to metabolic adjustments that occurred in all treatment groups (VE/VO₂ ALLO naive: 6.5±1.3; NT-shRNA: 6.7±1.7; PHOX2B-shRNA: 6.6±0.6; data not shown), suggesting that the hypoxic ventilatory response was not affected by PHOX2B knockdown.

## PHOX2B knockdown does not extend to C1 catecholaminergic neurons and does not affect mRNA expression of *Nmb* in RTN neurons

To exclude that the observed chemoreflex impairment was due to unintended knockdown of PHOX2B in adjacent areas of the brain, we quantified the total number of TH⁺/PHOX2B⁺ catecholaminergic C1 neurons in PHOX2B-shRNA rats compared to naive and NT-shRNA (*Figure 5A and B*). No differences were observed in either the TH⁺ cell numbers or in the TH⁺/PHOX2B⁺ neurons between the three treatment groups (TH⁺/PHOX2B⁺ cells: naive: 378.3±23.81; NT-shRNA: 391.7±33.08; PHOX2B-shRNA: 370.2±33.04 cells; *Figure 5B*; *Supplementary file 1*).

Neuromedin B is currently considered the most selective anatomical marker for chemosensitive RTN neurons (*Shi et al., 2017*) and was used in this study to identify RTN location, cell numbers and relative PHOX2B expression. Since changes in the level of the transcription factor PHOX2B could regulate the expression of its target genes, we investigated whether *Nmb* is a target gene of PHOX2B and it is affected by PHOX2B knockdown. We assessed *Nmb* expression in individual RTN cells of PHOX2B-shRNA rats and compared its cellular expression to both naive and NT-shRNA rats ( *Figure 5 C and D* ). Neuromedin B mRNA expression was calculated as total fluorescence intensity (CTCF) ratio in single cells of the RTN relative to the single cell *Nmb* expression measured at the level of NTS *Nmb* cells, which served as an internal control (*Figure 5D*). We observed no changes in relative fluorescence of *Nmb* in RTN neurons across the three groups (p=0.608), suggesting that *Nmb* levels in RTN neurons were not affected by PHOX2B knockdown.

To better analyse the effect of PHOX2B knockdown on the attenuation of the HCVR, we evaluated the correlation between the number of *Nmb*⁺/PHOX2B⁺ or Nmb⁺/PHOX2B⁻ cells in the RTN and the resulting HCVR using linear regression (*Figure 5E and F*). Our results indicate that there is a significant correlation between HCVR and number of Nmb⁺/PHOX2B⁺ ($r^2$=0.739 p<0.001; *Figure 5E*), and Nmb⁺/PHOX2B⁻ ($r^2$=0.482 p<0.001 *Figure 5F*) cells, suggesting that the number of PHOX2B expressing cells in the RTN is a good predictor of the chemoreflex response. Moreover, the nature of the correlation between PHOX2B⁻ cells and HCVR is the opposite of PHOX2B⁺, another important indicator that PHOX2B protein may contribute to the $CO_2$ sensing and consequently, the reduction of PHOX2B protein impairs the $CO_2$-chemoreflex.

## Gpr4 and Task2 mRNAs expression is affected by PHOX2B Knockdown

We next examined the mRNA expression of *Gpr4* and *Task2*, two important pH sensors for the central respiratory chemoreflex response in RTN neurons (*Figure 6A and D*; *Gestreau et al., 2010*; *Guyenet et al., 2016*; *Kumar et al., 2015*). As previously reported, the two pH sensors are expressed in the *Nmb* cells of the RTN in partially overlapping populations of chemosensory neurons (*Gestreau et al., 2010*; *Kumar et al., 2015*). In naive rats 91.6 ± 11.2% of *Nmb* cells were *Gpr4* positive, and 76.6 ± 14.8% of *Nmb* cells were also *Task2* positive (n=4 rats). These values were not significantly different in either NT-shRNA or PHOX2B-shRNA rats (NT-shRNA: 90.0 ± 15.7% *Nmb*⁺/*Gpr4*⁺ and + ± 21.8% *Nmb*⁺/*Gpr4*⁺, n=10 rats; PHOX2B-shRNA: 89.4 ± 18.3% *Nmb*⁺/*Gpr4*⁺ and + ± 27.8% *Nmb*⁺/*Gpr4*⁺, n=6 rats), suggesting that the fraction of *Nmb* cells expressing the two pH sensors were not affected by the PHOX2B shRNA treatment.

Because it is not yet known whether PHOX2B controls the expression of GPR4 AND TASK2, we quantified their single-cell mRNA expression (CTCF) in *Nmb* cells of the RTN. Averaged cell fluorescence for *Gpr4* and *Task2* mRNA in PHOX2B-shRNA rats was significantly reduced compared to naive and NT-shRNA rats (*Gpr4*: –34.4 ± 4.79% vs naive, –27.9 ± 5.71% vs NT-shRNA p<0.001; *Task2*: –39.0 ± 3.02% vs naive, –34 ± 2.62% vs NT-shRNA p<0.001; *Figure 6B and E*; *Supplementary file 1*). To better understand whether this reduction was ascribed to the specific loss of PHOX2B expression, we compared their single cell mRNA expression levels between *Nmb*⁺/PHOX2B⁻ and *Nmb*⁺/PHOX2B⁺ neurons in PHOX2B-shRNA rats. Both *Gpr4* and *Task2* mRNA expression was reduced in

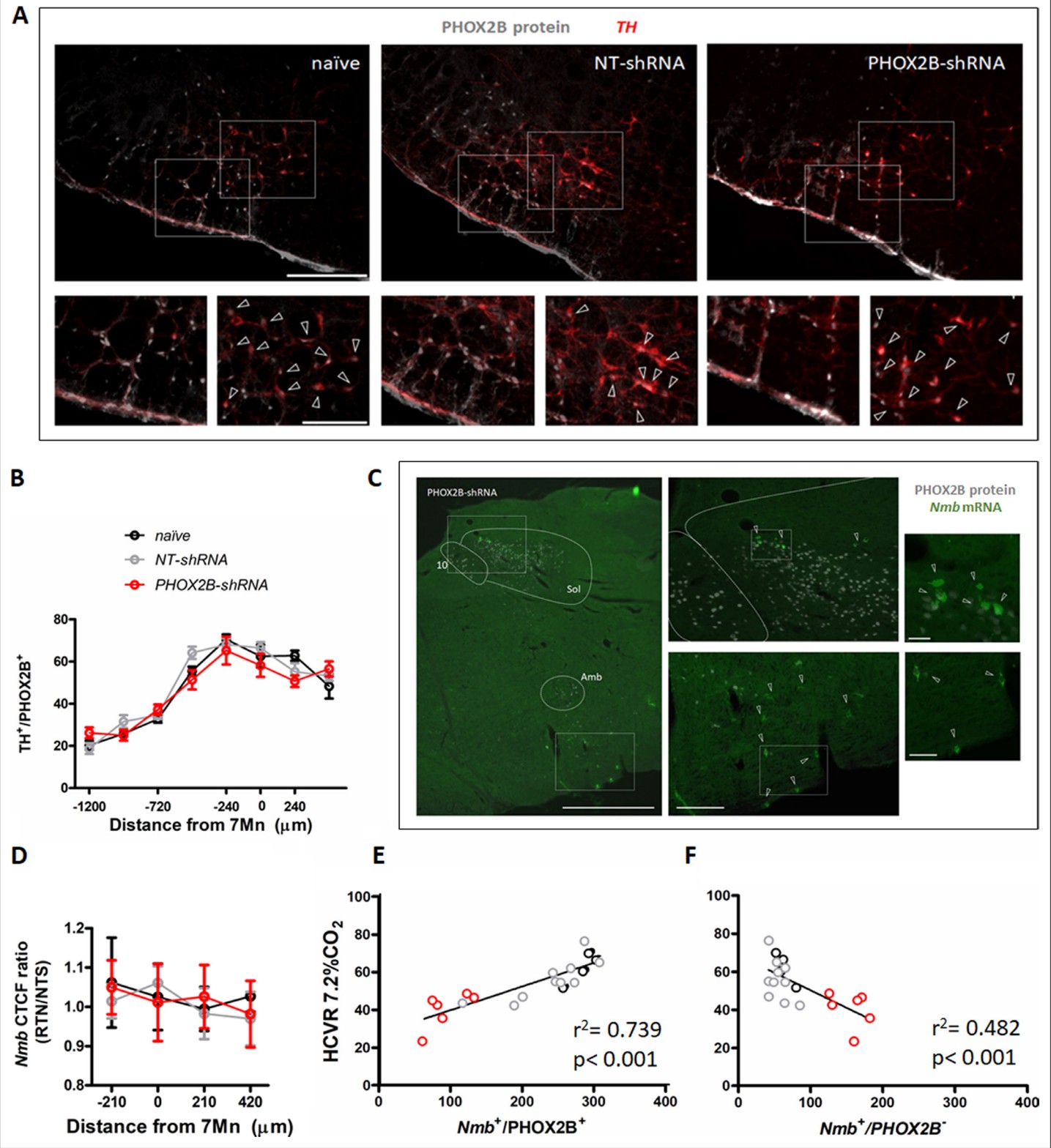

**Figure 5.** PHOX2B knockdown in RTN neurons does not alter TH or *Nmb* expression but impairs the hypercapnic ventilatory response. (**A**) PHOX2B protein (white) and TH (red) expression in C1 neurons of nave (left), NT-shRNA (middle), and PHOX2B-shRNA (right) rats. Magnified view at the bottom. Arrowheads indicate the colocalization of PHOX2B and TH protein in C1 neurones cells. Scale bar = 400 μm (top figures), 150 μm (bottom figures). (**B**) No differences were observed in the rostral-caudal distribution of TH+/PHOX2B+ catecholaminergic C1 neurons cells caudal to the RTN between naive (black, n=4), NT-shRNA (grey, n=10) and PHOX2B-shRNA (red, n=6) rats. (**C**) PHOX2B protein (white) and *Nmb* mRNA (green) expression in NT-shRNA

*Figure 5 continued on next page*

*Figure 5 continued*

rat at the level of NTS and RTN regions. Magnified view on the left. Arrowheads indicate cells with *Nmb* mRNA expression. Scale bar = 400 µm (right figures), 150 µm (left figures). (**D**) Quantification of single cells *Nmb* mRNA fluorescence intensity along the rostro-caudal extension of RTN in naive (black, n=4), NT-shRNA (grey, n=10) and PHOX2B-shRNA (red, n=6) rat calculated as average ratio between RTN and NTS cells showed no difference between treatment groups. Data are shown as average cell fluorescence value at different rostro-caudal levels. (**E-F**) X-Y plot of HCVR during 7.2% $CO_2$ exposure relative to the number of $Nmb^+/PHOX2B^+$ (E, slope is different from '0' at p<0.001; $r^2$=0.739) and $Nmb^+/PHOX2B^-$ (F, p<0.001 for difference between slopes; $r^2$=0.482) in the RTN.

The online version of this article includes the following source data for figure 5:

**Source data 1.** PHOX2B knockdown in RTN neurons does not alter TH or Nmb expression but impairs the hypercapnic ventilatory response.

$Nmb^+/PHOX2B^-$ cells compared to $Nmb^+/PHOX2B^+$ (*Gpr4*: –63.8 ± 3.9%; p=0.022; *Task2*: –63.2 ± 2.77%; p=0.029; *Figure 6C and F*). Our data suggest that PHOX2B knockdown not only causes a reduction in the HCVR but also a reduction in *Gpr4* and *Task2* mRNA levels in cells that are affected by PHOX2B knockdown.

## Discussion

The role of the transcription factor PHOX2B in the adult brain is still not fully understood. In this study, we reduced the expression of PHOX2B in the key chemosensory structure of the RTN with a selective viral shRNA approach to test whether a knockdown of PHOX2B expression in these neurons negatively impacts ventilation. Two weeks after viral infection, we observed a modest reduction of PHOX2B/*Nmb* expressing neurons in the RTN but no significant changes in basal $V_E$ or in the $CO_2$ chemoreflex. Four weeks after shRNA infection, we observed a further reduction in the expression of PHOX2B in *Nmb* neurons and a significant reduction of the $CO_2$ chemoreflex, while ventilation in normoxia or hypoxia was unaffected. Furthermore, the level of expression of *Gpr4* and *Task2* mRNAs, the two proton sensors responsible for the RTN neurons chemosensory properties were significantly reduced in RTN neurons that lacked PHOX2B protein expression, whereas *Nmb* mRNA expression level was unaffected by PHOX2B knockdown. These results suggest that PHOX2B reduction affects the transcriptional activity of *Nmb* neurons in the RTN and decreases the $CO_2$ response, possibly by reducing the expression of key proton sensors in the RTN.

### The role of PHOX2B in development and in the adult brain

PHOX2B is a transcription factor with an important role in the development and differentiation of neuronal structures in both the central and peripheral nervous system. Genetic loss of *Phox2b* expression in mice shows that PHOX2B is essential for the correct development of all autonomic ganglia in sympathetic, parasympathetic, and enteric nervous system and the distal ganglia of the facial (VII), glossopharyngeal (IX), and vagus (X) cranial nerves (*Pattyn et al., 1999*), in addition to carotid bodies and the solitary tract nucleus (*Dauger et al., 2003*). Furthermore, PHOX2B is involved in the formation of all branchial and visceral hindbrain motor neurons and its absence during the embryonic period impairs the development of motoneurons of the facial, trigeminal, ambiguus and dorsal motor nucleus of the vagal nerve. PHOX2B is also necessary for the specification and differentiation of the noradrenergic phenotype in multiple centres in the brain (*Brunet and Pattyn, 2002*; *Pattyn et al., 1999*).

Interestingly, the expression of PHOX2B persists into adulthood in selected neuronal populations, such as the ones involved in peripheral and central chemoreception, that is the carotid bodies, the solitary tract nucleus and the RTN, in addition to neurons in the dorsal motor nucleus of the vagus, the area postrema, the nucleus ambiguus, C1 catecholaminergic neurons and other scattered neurons throughout the brainstem (*Stornetta et al., 2006*). Such widespread expression within the adult brain suggests an important transcriptional role in neuronal survival and/or function. With the exception of its role in maintaining the noradrenergic phenotype, its function in the postnatal brain is unknown, in part because a full investigation of its transcriptional activity in neurons has not yet been performed.

Heterozygote polyalanine expansion mutations in the exon 3 of *PHOX2B* are responsible for the majority of cases of CCHS, a rare genetic disease that affect the autonomic nervous system and central $CO_2$ chemoreflexes (*Amiel et al., 2003*; *Di Lascio et al., 2020*; *Weese-Mayer et al., 2017*). While studies in transgenic mice expressing the mutated PHOX2B protein suggest that the RTN may not form or be severely impaired (*Dubreuil et al., 2008*; *Dubreuil et al., 2009*; *Ramanantsoa et al.,*

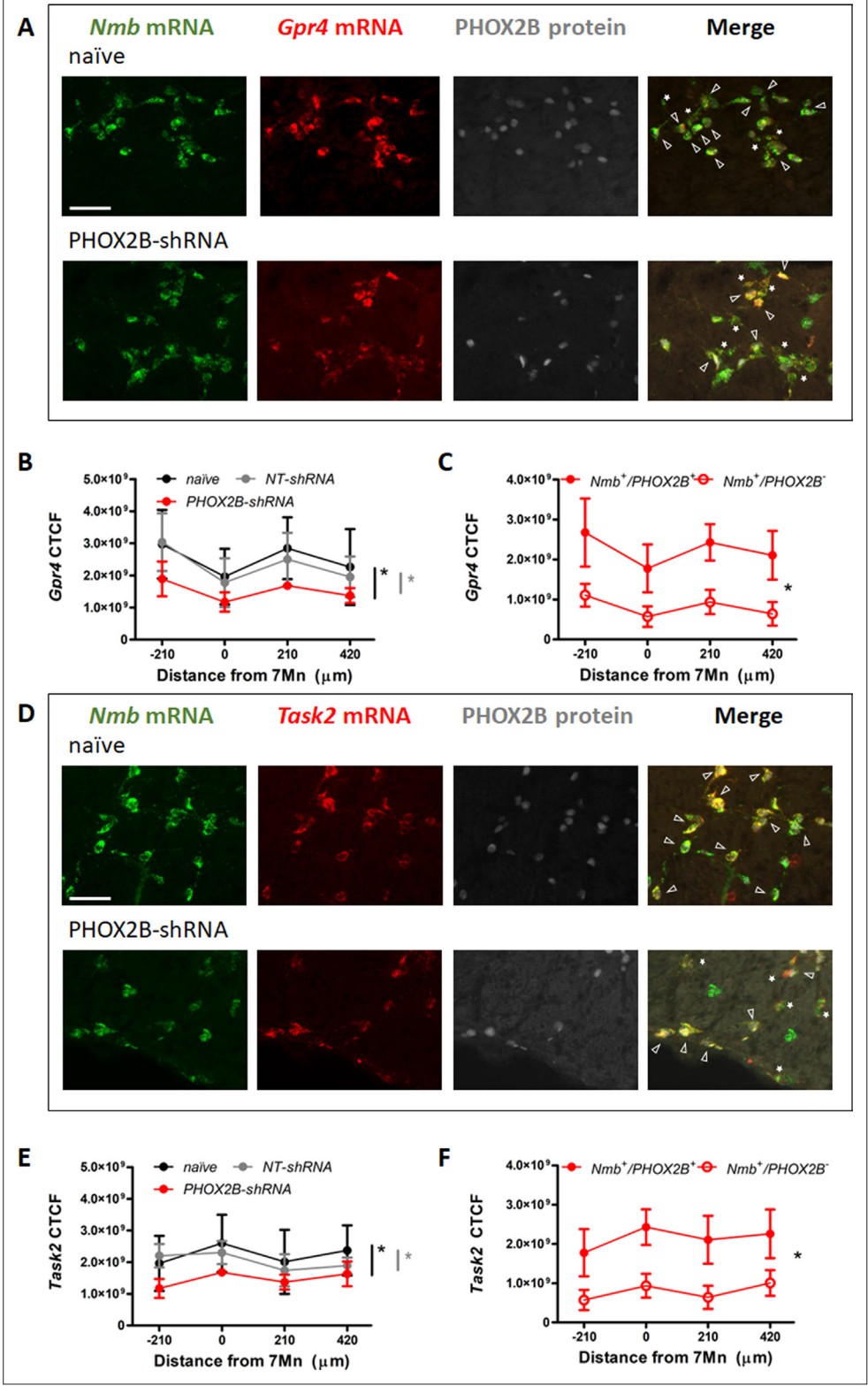

**Figure 6.** *Gpr4* and *Task2* mRNA expression within the RTN area in naive, and PHOX2B-shRNA injected rats
4 weeks post viral shRNA injection. (**A, D**) *Nmb* (green), *Gpr4*, *Task2* mRNA (red) and PHOX2B protein (grey)
expression in RTN neurons in naive (top) and PHOX2B-shRNA (bottom) rats. Scale bar = 150 µm. Arrowheads
indicate colocalization of *Gpr4* (**A**) and *Task2* (**D**) with *Nmb*+/PHOX2B+ neurons. Asterisks indicate colocalization
of *Gpr4* (**A**) and *Task2* (**D**) with *Nmb*+/PHOX2B- neurons. (**B, E**) Quantification of single cells *Gpr4* and *Task2*

*Figure 6 continued on next page*

*Figure 6 continued*

mRNA fluorescence intensity along the rostro-caudal extension of RTN in naive (black), NT-shRNA (grey) and PHOX2B-shRNA (red) rats. Black*, different from naive, p<0.001. Grey*, different from NT-shRNA, p<0.001 (**C, F**) Quantification of single cells *Gpr4* and *Task2* mRNA fluorescence staining intensity along the rostro-caudal extension of RTN in Nmb$^+$/PHOX2B$^+$ (red filled dot) and Nmb$^+$/PHOX2B$^-$ (red empty dot) neurons in PHOX2B-shRNA rats (C, p=0.022); (F, p=0.029). Data are shown as average cell fluorescence value at different rostro-caudal levels of the RTN. Mean corrected total cell fluorescence (CTCF) value ± SEM combined (naive, n=4; NT-shRNA, n=10; PHOX2B-shRNA, n=6) (see Materials and methods for detail). One-way ANOVA repeated measures.

The online version of this article includes the following source data for figure 6:

**Source data 1.** Gpr4 and Task2 mRNA expression within the RTN area in naive, and PHOX2B-shRNA injected rats 4 weeks post viral shRNA injection.

*2011*), the function of the wild-type protein in these neurons has never been tested beyond the post-natal period. Thus, in order to determine whether PHOX2B is necessary for the survival and/or chemo-sensory function of adult PHOX2B$^+$/*Nmb*$^+$ RTN neurons in vivo, we used a shRNA viral approach to progressively knockdown the PHOX2B protein in these neurons.

## Modest reduction of PHOX2B in RTN does not impair ventilation

Although the PHOX2B shRNA approach was not selective for *Nmb* neurons, we used small volumes and localized injections to target the RTN. We identified RTN neurons by the mRNA expression of the neuropeptide NMB (*Shi et al., 2017*) and observed a reduced number in both total *Nmb* and *Nmb*$^+$/PHOX2B$^+$ neurons in PHOX2B shRNA-treated rats compared to naive and NT-shRNA rats two weeks after viral infection. Interestingly, despite the small reduction in total *Nmb* cells of the RTN in both NT-shRNA and PHOX2B-shRNA injected rats, possibly due to some off-target effects due to activation of innate immunity or saturation of the microRNA pathway (*McBride et al., 2008*; *van Gestel et al., 2014*), no significant impairment in ventilation was observed. Even though PHOX2B may be critical for chemosensory function in RTN neurons, the lack of significant ventilatory impairment with a modest reduction of PHOX2B expressing neurons is not surprising. In our previous study (and the ones of others), in which RTN chemosensory neurons were lesioned with Substance P saporin toxin, $CO_2$-chemoreflex impairment became significant (~30% $V_E$ Allo reduction vs. pre-surgical baseline) only when *Nmb* neurons of the RTN (*Nmb*$^+$/PHOX2B$^+$ and *Nmb*$^+$/PHOX2B$^-$ neurons) were reduced by ~63% compared to naive controls (medium lesion range: 50–81% loss of *Nmb* neurons; *Janes et al., 2024*; *Souza et al., 2023*). Therefore, even though loss of PHOX2B severely altered the chemo-sensitive function in a fraction of *Nmb* cells, we would not expect significant changes in ventilation only with a<50% RTN cell impairment or loss.

## Four weeks of PHOX2B knockdown impairs central $CO_2$ chemoreception

Four weeks post- shRNA viral injection, the fraction of *Nmb*$^+$ cells expressing PHOX2B was further reduced by 67% compared to naive and by 61% compared to NT-shRNA rats. PHOX2B knockdown was also restricted to RTN neurons, as adjacent C1 TH$^+$ neurons did not show any change in number of TH$^+$/PHOX2B$^+$ expressing cells, although we cannot exclude that some C1 cells may have been infected, and their relative PHOX2B expression levels were slightly reduced. To support the lack of significant alterations associated with the possible loss of C1 function was the lack of significant changes in the hypoxic response that has been shown to be dependent on C1 neurons (*Malheiros-Lima et al., 2017*).

Although we did not specifically quantify the relative mRNA expression change in intracellular *Phox2b* levels within infected neurons, we measured the success of PHOX2B knockdown by assessing the overall change in the number of *Nmb* cells that had no detectable levels of PHOX2B protein in the nuclei. Thus, it is possible that this method led us to underestimate the success of the overall PHOX2B shRNA knockdown in the RTN, as some RTN neurons may still have detectable levels of PHOX2B protein, albeit decreased.

In PHOX2B-shRNA rats, we did not observe any respiratory function changes in room air or in hypoxia, similar to RTN lesioning studies (*Janes et al., 2024*; *Souza et al., 2023*; *Guyenet et al., 2019*). When tested in hypercapnia though, PHOX2B-shRNA rats displayed a reduced HCVR

compared to both baseline (i.e. pre-surgery), naive and NT-shRNA rats. The impairment of $V_E$ Allo was primarily a result of blunted $V_T$, and analysis of the convective exchange ratio (i.e. $V_E/VO_2$ ALLO) confirmed hypoventilation in 5% and 7.2% $CO_2$ only for PHOX2B-shRNA rats, suggesting a specific impairment of chemosensitive properties in RTN neurons with PHOX2B knockdown. These results are in line with previous studies in which medium RTN lesions (i.e. loss of >60% $Nmb^+$ cells) elicited by saporin toxin injection resulted in a significant $CO_2$-chemoreflex impairment (*Janes et al., 2024*; *Souza et al., 2018*). Hence, killing RTN neurons or knocking down PHOX2B in *Nmb* cells of the RTN gave comparable results.

Changes in the $CO_2$ response following 4 weeks of PHOX2B-shRNA treatment could occur because the reduced expression of the transcription factor PHOX2B causes changes in the transcriptional machinery of RTN neurons and alters transcription of $CO_2$/pH sensing proteins or other key elements for their chemosensitive function. These transcriptional changes could also affect neuronal survival. In fact, a small but significant loss of NMB expressing RTN neurons may contribute to the reduction in the $CO_2$ response, although multiple studies have demonstrated that significant cell loss must occur in the RTN in order to show a respiratory function impairment (*Janes et al., 2024*; *Souza et al., 2018*; *Nattie and Li, 2002*; *Takakura et al., 2008*; *Takakura et al., 2014*).

In our experiments, we observed a 35% reduction in total $Nmb^+$ cells compared to naive rats and 23% compared to NT-shRNA, possibly due to cell death within the first 14 days post-infection. Interestingly the loss of *Nmb* cells did not increase with time (although the fraction of $Nmb^+$/PHOX2B$^+$ neurons decreased) and the central chemoreflex at 2 weeks post-surgery was not impaired, suggesting that: (i) the observed reduction in overall *Nmb* cell number in RTN occurring in the first 2 weeks is not responsible for the $CO_2$ chemoreflex impairment that emerges 4 weeks post-infection; (ii) PHOX2B expression is most likely not necessary for neuronal survival of adult *Nmb* neurons of the RTN; (iii) a large reduction in *Nmb* neurons expressing PHOX2B$^+$ is necessary to impair the HCVR (as we observed at 4 weeks). The interpretation that PHOX2B expression contributes positively to the chemoreflex is further supported by our linear regression analysis showing that $Nmb^+$/PHOX2B$^+$ cell number was the best predictor of the HCVR ($r^2=0.739$). The initial cell loss, which we observed also in NT-shRNA, and more prominently with larger injection volume of virus, may be attributed to some inherent cell death associated with the surgical procedure (although not observed in similar studies performed by the same investigators, *Janes et al., 2024*) or, more likely, with the potential off-target toxic effects associated with shRNA procedures (*McBride et al., 2008*; *van Gestel et al., 2014*).

## PHOX2B knockdown reduces mRNA expression of proton sensors in the Nmb cells of the RTN

Current theories on central respiratory chemosensitivity postulate that changes in $CO_2$/pH in the RTN are detected through two pH sensors, the proton-activated receptor GPR4 and the pH-sensitive K$^+$ channel TASK2 that are expressed in partially overlapping populations of NMB expressing RTN neurons (*Kumar et al., 2015*). Because it is not known whether PHOX2B has any influence on the expression of TASK2, GPR4 or even the expression of the neuropeptide NMB used to anatomically identify RTN neurons, we quantified their mRNA at cellular levels (calculated as CFTC) and observed no changes in relative fluorescence of *Nmb* mRNA in the RTN relative to *Nmb* levels in unrelated cells in the NTS, suggesting that *Nmb* expression is not affected by PHOX2B shRNA knockdown and that *Nmb* is most likely not a target gene of PHOX2B in adult RTN neurons.

Furthermore, we determined the mRNA expression levels of *Task2* and *Gpr4* in both $Nmb^+$/PHOX2B$^+$ and Nmb$^+$/PHOX2B$^-$ neurons in shRNA rats. Our results indicate that although the fraction of *Nmb* cells of the RTN expressing *Gpr4* and *Task2* did not change across treatment, the levels of *Gpr4* and *Task2* in *Nmb* neurons of PHOX2B-shRNA rats was reduced compared to the naive and NT-shRNA groups. Furthermore, there was a significant reduction of both *Gpr4* and *Task2* levels within $Nmb^+$/PHOX2B$^-$ cells compared to $Nmb^+$/PHOX2B$^+$ cells. Because no good antibodies are currently available to detect protein levels of either NMB, GPR4 or TASK2, our results are only based on changes in mRNA levels, thus we can only speculate that a reduction in *Gpr4* and *Task2* mRNA would translate in a reduction in the protein levels and consequent reduction of RTN neurons chemosensitive properties. Direct electrophysiological recordings in RTN neurons will be able to address the changes in $CO_2$/pH sensitivity of RTN neurons at cellular level.

## PHOX2B knockdown may impair CO$_2$-sensing through additional transcriptional targets

Since loss of PHOX2B reduces expression of TH and dopamine beta hydroxylase enzymes in catecholaminergic neurons in vivo and in vitro (*Fan et al., 2011*), it is possible that in NMB cells of the RTN, reduction of PHOX2B alters the transcriptional machinery of other proteins (in addition to GPR4 and TASK2) and impairs their function in central chemoreception. For example, a change in the expression of enzymes, neurotransmitters, receptors, and ion channels in RTN neurons could affect the excitatory signal transmission to preBötzinger Complex and the respiratory network specifically during CO$_2$ challenges (*Guyenet et al., 2016*). Further studies will be needed to address and identify additional transcriptional changes induced by PHOX2B knockdown in vivo.

An interesting observation from our histological data was the reduction in the overall number of *Nmb* cells at 2- and 4-weeks following PHOX2B shRNA viral injections. Although some cell death may be associated with either surgical procedures or off-target effects associated with the shRNA approach (*McBride et al., 2008*; *van Gestel et al., 2014*), it is intriguing to speculate that loss of PHOX2B expression may have some effects on viability of RTN neurons in vivo. Even though this is a possibility, especially with an even more severe knockdown, we did observe a large proportion of *Nmb* cells devoid of PHOX2B protein expression, suggesting that absence of PHOX2B is compatible with neuronal survival, at least in our experimental time frame.

## Relevance to CCHS and its pathogenic mechanisms

Our results contribute to further our understanding on potential pathogenic mechanisms in CCHS (*Di Lascio et al., 2018a*). Homozygous knockout mice for PHOX2B die during gestation (*Pattyn et al., 1999*; *Pattyn et al., 2000*) demonstrating a key role of this transcription factor in cellular proliferation, migration and differentiation during embryonic development, whereas heterozygotes for PHOX2B survive birth and live into adulthood, but display apneas (*Durand et al., 2005*) and impairment of the hypoxic and hypercapnic ventilatory responses in the neonatal period (*Dauger et al., 2003*). Here we show that, independent of the PHOX2B role during development, PHOX2B is still required to maintain proper CO$_2$ chemoreflex responses in the adult brain.

Heterologous expression of the mutant PHOX2B alters development of RTN in mice and impairs their chemoreflex response (*Dubreuil et al., 2008*; *Dubreuil et al., 2009*). The effects of the expression of the mutant PHOX2B on wild-type PHOX2B protein and its transcriptional activity is not known in vivo yet, although in vitro data indicates that the mutated PHOX2B protein alters wild-type PHOX2B conformation and its ability to form homo- and hetero-dimers (*Di Lascio et al., 2016*; *Trochet et al., 2005*; *Trochet et al., 2008*), its affinity for DNA and coactivators, as well as its degradation rate (*Nagashimada et al., 2012*; *Wu et al., 2009*). It has also been shown that mutant PHOX2B causes the formation of cytoplasmic aggregates (*Bachetti et al., 2005*; *Trochet et al., 2005*; *Trochet et al., 2008*) and fibrils in vitro (*Pirone et al., 2019*), and dysregulates the transcriptional function of wild-type PHOX2B (*Di Lascio et al., 2013*; *Parodi et al., 2012*), thus changing expression of important target genes such as DBH, PHOX2A and TLX2 (*Bachetti et al., 2005*; *Di Lascio et al., 2013*; *Trochet et al., 2005*). Based on this evidence, different CCHS pathogenic mechanisms have been proposed (e.g. PHOX2B loss-of-function, dominant-negative or toxic function of the mutant protein) including gene- and cell-specific transcriptional dysregulation (*Di Lascio et al., 2018a*; *Di Lascio et al., 2020*). Interestingly, only a handful of PHOX2B target genes have been identified and the function of PHOX2B, and its mutated forms in respiratory control and CCHS pathogenesis is still under investigation.

Our data suggest that, in addition to potential effect of mutated PHOX2B on development and cellular function in CCHS pathogenesis, the expression of wild-type PHOX2B has an important role in respiratory control that extends the developmental period and its reduction in CCHS may contribute to the respiratory impairment in this disorder.

## Materials and methods

Experiments were performed using male Sprague–Dawley rats (starting weight range 250–350 g) born in the University of Alberta animal facility to pregnant dams obtained from Charles River (strain code: 001, Senneville, QC). Rats were housed at the University of Alberta Health Sciences Animal Housing Facility and maintained on a 12 hr dark/light cycle with food and water available ad libitum.

Handling and experimental procedures were approved by the Health Science Animal Policy and Welfare Committee at the University of Alberta (AUP#461) and performed in accordance with guidelines established by the Canadian Council on Animal Care.

## ShRNA viral injection in the retrotrapezoid nucleus

In order to knockdown Phox2b expression at the level of RTN neurons, we used a mix of two shRNA clones targeting two different sequences of the Phox2b mRNA carried by a non-replicating lentivirus vector ($1\times10^9$ VP/ml; TRCN000041283: GCCTTAGTGAAGAGCAGTATG and TRCN0000096437: CCTC TGCCTACGAGTCCTGTA; Sigma Aldrich, St. Louis, MA, USA). The shRNAs were designed against the Phox2b mouse sequence (Ref Seq NM_008888) which shares 100% homology with the Phox2b rat sequence (Ref Seq XM_008770167).

Metacam analgesic was administered 1 hr prior to surgery (2 mg/kg) to reduce post-surgical pain. Rats were anesthetized with a ketamine/xylazine mixture (100 mg/kg+10 mg/kg, respectively; i.p.); the anaesthetic level was assessed by lack of paw pinch reflex and maintenance of a regular breathing rate. Additional anaesthesia was administered as needed. The rat was positioned on a stereotactic apparatus (Kopf Instruments, Tujunga, CA, USA) and a total of four microinjections (200 or 100 nL per injection; two rostro-caudally aligned injections per side) were made lateral to the midline and ventral to the facial motor nucleus under aseptic conditions. Coordinates were as follows in mm from obex (medio-lateral/rostro-caudal/dorso-ventral): ±1.8, +2.0, –3.5; ±1.8, +2.4, –3.6 (*Paxinos and Watson, 2007*). Injections were made using a glass microelectrode (30 µm diameter tip, Drummond Scientific, PA, USA). Twenty-five rats (8 rats 200 nl/injection; 17 rats 100 nl/injection) underwent shRNA viral injection (PHOX2B-shRNA) and 23 rats (6 rats 200 nl/injection; 17 rats 100 nl/injection) received a non-target shRNA viral injection (NT-shRNA; SHC016: MISSION pLKO.1-puro non-Target shRNA Control Plasmid DNA; Sigma-Aldrich, St. Louis, MA, USA) to determine any effects of surgery and viral constructs on respiratory behaviour. Rats were randomly assigned to treatment group. Eight naive rats, receiving no surgery and maintained in the same housing conditions, were used as additional controls for both respiratory function and anatomical experiments. Following each injection, the glass electrodes were left in place for 3–5 min to minimize backflow of virus up the electrode track. At the end of the surgery, the incision was sutured, and rats received local anaesthetic bupivacaine (0.1 mL, s.c.) and were treated with metacam analgesic for 72 hr. Rats recovered for 7 days before the next respiratory function testing.

## Data acquisition and analysis of respiratory measurements

Rats were habituated to whole-body plethysmography chambers (Buxco, 5 L) once, 3–4 days before baseline recordings. On the day of the experiment, rats were placed in the chamber and testing commenced once the animals were quietly awake (typically 20–30 min). Gas mixtures were delivered at 1.5 L/min using a GSM-3 (CWE Inc, Ardmore, PA, USA) and monitored using a Gas Analyzer (AD Instruments, Sydney, Australia). Ventilatory parameters were measured during exposure to different air compositions for 8–15 min each. Room air: 21%$O_2$+0%$CO_2$ (balanced with $N_2$); Normoxic Hypercapnia: 21%$O_2$+5%$CO_2$; 21%$O_2$+7.2%$CO_2$; Normocapnic Hypoxia: 10.6%$O_2$+0% $CO_2$. The order of chemoreflex was randomized from one session to the next.

As previously described (*Cardani et al., 2022*; *Janes et al., 2024*) respiratory data were analysed from rats during quiet wakefulness in the last 5 min period of each gas composition using the barometric method (open-flow system; *Cardani et al., 2022*; *Janes et al., 2024*; *Mortola and Frappell, 1998*; *Seifert et al., 2000*). Raw pressure signals were acquired with a Validyne differential pressure transducer connected to a CD15 carrier demodulator (Validyne Engineering, Northridge, CA, USA) and digitized using a Powerlab 8/35 (AD Instruments, Sydney, Australia). Analysis was done using Labchart 8 (v8.1.19, AD Instruments, Colorado Springs, CO, USA) to determine tidal volume ($V_T$) and breathing frequency ($f_R$). The amplitude of the pressure signal was converted to $V_T$ (ml·$kg^{-1}$) using the equations of *Drorbaugh and Fenn, 1955* (*Drorbaugh and Fenn, 1955*) and calibrated against 1 mL of dry air injected into the empty chamber using a rodent ventilator (Harvard Rodent Ventilator Model 683, Holliston, MA, USA $f$=50, 75, 100, 150 beats·$min^{-1}$). Humidity and chamber temperature were monitored using a RH-300 water vapour analyzer (Sable Systems, Las Vegas, NV, USA) and a HPR Plus Handheld PIT Tag reader (Biomark, Boise, ID, USA) respectively, depending on the setup. Rectal temperature was read through the temperature probes of a Homeothermic Monitor

(507222 F, Harvard Apparatus, Holliston, MA, USA). Minute ventilation ($V_E$) was calculated as $f_R$ x $V_T$ and expressed as ml·min$^{-1}$·kg$^{-1}$. Rats gained, on average, 160±4 g over the experimental paradigm (naive: 158±19 g; NT-shRNA: 202±25 g; PHOX2B-shRNA: 133±7 g); we therefore applied an allometric correction to $V_E$ to account for non-linear effects of weight gain ($V_E$ ALLO) (*Mortola et al., 1994*).

Metabolic parameters $VO_2$ and $V_E/VO_2$ were calculated by pull-mode indirect calorimetry and allometrically scaled to be consistent with ventilation (*Mortola et al., 1994*). The % composition of dry gas flowing into, and out of the recording chamber was measured by using an AD Instruments Gas Analyzer (Colorado Springs, CO) and applying the equations of Depocas and Hart (*Depocas and Sanford Hart, 1957*) and *Lighton, 2021*.

Weight measurements were taken prior to every recording session. Rats breathing function was measured the week prior to surgery (baseline recording) and weekly following shRNA injections to determine the time course of ventilatory impairment.

Respiratory data were compared for naive, NT-shRNA and PHOX2B-shRNA pre- and post-injection using a mixed factor ANOVA (independent factor = treatment, repeated measures factor = pre and post-lesion; *Figures 1 and 4*). Post-hoc analysis used the Bonferroni correction factor, with $p<0.05$ considered to be significant. Ventilatory data are reported as mean ± standard deviation (SD).

## In Situ hybridization (RNAScope) and immunofluorescence

Either 2 or 4 weeks after viral injection, rats were transcardially perfused with saline 0.9% NaCl followed by 4% paraformaldehyde (PFA) and the brains were post-fixed in 4% PFA overnight and cryoprotected in 30% sucrose in 1 X Phosphate Buffered Saline (PBS). Brains were then frozen in O.C.T compound (Fisher Scientific) and sectioned on a cryostat (MODEL CM1950, Leica Biosystems, Buffalo grove, IL, USA) at 30 µm and stored in cryoprotectant buffer at –20 °C until processing. Sections were mounted on slides for combined RNAScope in situ hybridization (Advanced Cell Diagnostics-ACD Bio, Newark CA, USA) and immunofluorescence assay and processed as previously detailed (*Biancardi et al., 2021*; *Cardani et al., 2022*). The mRNA expression for *Neuromedin B* (*Nmb*) was used as marker of RTN $CO_2$-sensing neurons (*Shi et al., 2017*). Slides were incubated with probes for *Nmb* (Rn-NMB-C2 #494791-C2, ACDBio, Newark, CA, USA), *Phox2b* (Rn-Phox2b-O1-C1 #1064121-C1, ACDBio), G-protein-coupled receptor 4 (*Gpr4*) (Mm-Gpr4-C1, #427941, ACDBio), and potassium channel, subfamily K, member 5 (*Kcnk5* or *Task-2*) (Mm-Kcnk5-C3, #427951-C3, ACDBio) for 2 hr at 40 °C. Gpr4 and Kcnk5 probes are design against the mouse mRNA sequence (Ref Seq NM_175668.4 and NM_021542.4, respectively). However, the alignment of the mouse mRNA sequences with those of rat (GPR4: Ref Seq NM_001025680.1; TASK2: Ref Seq NM_1039516.2) showed an identity of 94% and 96%, respectively, at the level of the target region (Base Pairs 1030–150) (BLAST program, National Library of Medicine, National Center for Biotechnology Information, NIH). In parallel, two sections/rat were treated with positive (low copy housekeeping gene), and negative (non-specific bacterial gene) control probes provided by ACDBio. Finally, slides were processed using the RNAScope Multiplex Fluorescent Assay kit V2 (ACDBio) according to the manufacturer's instructions. The probes were visualized using Opal 520 and Opal 570 reagent (1:500 and 1:1000, respectively; PerkinElmer, Woodbridge, ON, CA). PHOX2B immunoreactivity was detected using mouse monoclonal PHOX2B antibody (B-11: sc-376997, 1:100, Santa Cruz Biotechnology, RRID: AB_2813765) incubated overnight in 0.3% TritonX-100, 1%NDS in PBS followed by donkey CY5-conjugated anti-mouse IgG (1:200; Jackson Immuno Research Laboratories Inc) in PBS +1% NDS for 2 hr. Slides were then washed in PBS and cover-slipped with Fluorosave mounting media (EMD Millipore).

We also performed staining for tyrosine hydroxylase (TH) to identify and quantify C1 cells (TH$^+$/PHOX2B$^+$) following shRNA injection. Briefly, perfusion and tissue fixation/freezing were done as described above. Floating sections were stained with rabbit anti TH antibody (AB152,1:1000, Millipore SIGMA, USA, RRID:AB_390204), and mouse anti PHOX2B (B-11: sc-376997, 1:100, Santa Cruz Biotechnology, RRID: AB_2813765). Antibodies were visualized using donkey CY3 (TH) and CY5 (PHOX2B) conjugated IgG (1:200; Jackson Immuno Research Laboratories Inc). The sections were washed in PBS and mounted on slides and cover-slipped with Fluorosave mounting media.

## Cell counting, imaging and data analysis

To quantify *Nmb*+/PHOX2B- and *Nmb*+/PHOX2B+neurons within the RTN region, we analysed one every seven sections (210 µm interval; 8 sections/rat in total) along the rostrocaudal distribution of

the RTN on the ventral surface of the brainstem and compared total bilateral cell counts of PHOX2B-shRNA rats with non-target control (NT-shRNA) and naive rats. Cells that expressed *Nmb* and *Phox2b* mRNAs but did not show co-localization with PHOX2B protein were considered *Nmb*+/PHOX2B-.

The Corrected Total Cell Fluorescence (CTCF) signal for *Nmb*, *Gpr4* and *Task2* mRNAs was quantified as previously described (*Cardani et al., 2022*; *McCloy et al., 2014*). Briefly, a Leica TCS SP5 (B-120G) Laser Scanning Confocal microscope was used to acquire images of the tissue. Exposure time and acquisition parameters were set for the naive group and kept unchanged for the entire dataset acquisition. The collected images were then analysed by selecting a single cell at a time and measuring the area, integrated density and mean grey value (*McCloy et al., 2014*). For each image, three background areas were used to normalize against autofluorescence. We used 4 sections/rat (210 μm interval) to count *Nmb*, *Gpr4* and *Task2* mRNA CTCF in the core of the RTN area where several *Nmb* cells could be identified. For each section, two images were acquired with a 20×objective, so that at least fifty cells per tissue sample were obtained for the mRNA quantification analysis. To evaluate changes in *Nmb* mRNA expression levels following PHOX2B knockdown at the level of the RTN, we compared, the fluorescence intensity of each RTN *Nmb* cell (223.2±37.1 cells/animal) with the average fluorescent signal of *Nmb* cells located dorsally in the NTS (4.3±1.2 cells/animal) (*Nmb* CTCF ratio RTN/NTS) as we reasoned that the latter would not be affected by the shRNA infection and knockdown.

To quantify *Gpr4* and *Task2* mRNA expression in *Nmb* cells of the RTN, we first quantified single-cell CTCF for either *Gpr4* (200.7±13.2 cells/rat) or *Task2* (169.6±10.3 cells/rat) mRNA in *Nmb* cells of the RTN in the three experimental groups (naive, NT shRNA and PHOX2B shRNA) independent of their PHOX2B expression. We then compared CTCF values of *Gpr4* and *Task2* mRNA between *Nmb*+/PHOX2B+ and *Nmb*+/PHOX2B- RTN neurons in PHOX2B-shRNA rats to address changes in their mRNA expression induced by PHOX2B knockdown.

To assess whether shRNA knockdown affected either the number or the expression of PHOX2B in TH expressing C1 neurons, we counted the number of TH+/PHOX2B+ and TH+/PHOX2B- cells along the ventrolateral medulla (8 sections/rat; 240 μm interval) using an EVOS fluorescent microscope (Thermo Fisher Scientific).

Digital colour photomicrographs were acquired using a Leica TCS SP5 (B-120G) Laser Scanning Confocal microscope. Image J (version 1.54 f; National Institutes of Health, Bethesda, MD, USA) and Excel (Microsoft Office 365 for Windows) were used for cell counting and the measurements of fluorescence intensity. Statistical analysis of cell counts, and fluorescent signals was made using one-way ANOVA (*Figures 2C and 3C*, *Figure 1—figure supplement 1*) and repeated-measures ANOVA (*Figures 5B, D , and 6*) with GraphPad Prism 8 Software (GraphPad Software, Inc, San Diego, CA, USA). Linear regressions to determine the effect of cell counts (*Nmb*+/PHOX2B+, *Nmb*+/PHOX2B-, total cells) on chemoreflex magnitude were run using a combined dataset for naive, NT-shRNA and PHOX2B-shRNA rats (SPSS version 13.0; *Figure 5E–F*). p values of<0.05 were considered significant and data are reported as mean ± standard deviation (SD).

## Acknowledgements

SC is supported by the Canadian Lung Association BaO Fellowship. TAJ is supported by the Canadian Institutes for Health Research (CIHR) Postdoctoral Fellowships. SP is supported by a Women and Children's Health Research Institute Innovation Grant and CIHR Project scheme grant.

Confocal images were acquired at the University of Alberta Faculty of Medicine & Dentistry Cell Imaging Core, RRID:SCR_019200, which receives financial support from the Faculty of Medicine & Dentistry, the University Hospital Foundation, Striving for Pandemic Preparedness – The Alberta Research Consortium, and Canada Foundation for Innovation (CFI) awards to contributing investigators

## Additional information

### Funding

| Funder | Grant reference number | Author |
|---|---|---|
| Canadian Institutes of Health Research | Project Scheme Grant 486764 | Silvia Pagliardini |
| Women and Children Health Research Institute | Innovation Grant | Silvia Pagliardini |
| Canadian Lung Association | Post-doctoral Fellowship | Silvia Cardani |
| Canadian Institutes of Health Research | Post-doctoral Fellowship | Tara A Janes |

The funders had no role in study design, data collection and interpretation, or the decision to submit the work for publication.

### Author contributions

Silvia Cardani, Conceptualization, Data curation, Formal analysis, Validation, Investigation, Visualization, Methodology, Writing – original draft, Writing – review and editing; Tara A Janes, Conceptualization, Data curation, Formal analysis, Validation, Visualization, Methodology, Writing – review and editing; William Betzner, Data curation, Formal analysis, Validation, Investigation, Methodology, Writing – review and editing; Silvia Pagliardini, Conceptualization, Resources, Data curation, Supervision, Funding acquisition, Validation, Investigation, Visualization, Methodology, Writing – original draft, Project administration, Writing – review and editing

### Author ORCIDs

Silvia Cardani ⓘ https://orcid.org/0000-0001-9758-2649
Tara A Janes ⓘ http://orcid.org/0000-0002-2824-9487
William Betzner ⓘ https://orcid.org/0000-0002-9071-0705
Silvia Pagliardini ⓘ https://orcid.org/0000-0002-1482-9173

### Ethics

Experiments were performed using male Sprague-Dawley rats housed at the University of Alberta Health Sciences Animal Housing Facility. Handling and experimental procedures were approved by the Health Science Animal Policy and Welfare Committee at the University of Alberta (AUP#461) and performed in accordance with guidelines established by the Canadian Council on Animal Care.

Reviewer #1 (Public Review): https://doi.org/10.7554/eLife.94653.3.sa1
Reviewer #2 (Public Review): https://doi.org/10.7554/eLife.94653.3.sa2
Reviewer #3 (Public Review): https://doi.org/10.7554/eLife.94653.3.sa3
Author response https://doi.org/10.7554/eLife.94653.3.sa4

## Additional files

### Supplementary files

• Supplementary file 1. Details on the main statistical analyses reported in the indicated figures.
• MDAR checklist

### Data availability

All data generated or analysed during this study are included in the manuscript and supporting files.

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
