## [Editor Report · eLife assessment]

This **important** study utilizes a viral-mediated short hairpin RNA (shRNA) approach to investigate in a novel way the role of the wild-type PHOX2B transcription factor expressed in critical chemosensory neurons in the brainstem retrotrapezoid nucleus (RTN) region for maintaining normal CO2 chemoreflex control of breathing in adult rats. The **convincing** results show blunted ventilation during elevated inhaled CO2 (hypercapnia) with knockdown of PHOX2B, accompanied by a reduced expression of Gpr4 and Task2 mRNA for the proposed RTN neuron proton sensor proteins GPR4 and TASK2. These results indicate that maintained expression of wild-type PHOX2B affects respiratory control in adult animals, complementing previous studies showing that PHOX2B-expressing RTN neurons may be critical for chemosensory control throughout the lifespan, and with implications for neurological disorders involving the RTN, which will be of interest to neuroscientists studying respiratory neurobiology as well as the neurodevelopmental control of motor behavior.

---

## [Referee Report · Reviewer #1 (Public Review)]

Summary:

This important study investigated the role of the PHOX2B transcription factor in neurons in the key brainstem chemosensory structure, the retrotrapezoid nucleus (RTN), for maintaining proper CO2 chemoreflex responses of breathing in the adult rat in vivo. PHOX2B has an important transcriptional role in neuronal survival and/or function, and mutations of PHOX2B severely impair the development and function of the autonomic nervous system and RTN, resulting in the developmental genetic disease congenital central hypoventilation syndrome (CCHS) in neonates, where the RTN may not form and is functionally impaired. The function of the wild-type PHOX2B protein in adult RTN neurons that continue to express PHOX2B is unknown. By utilizing a viral PHOX2B-shRNA approach for the knockdown of PHOX2B specifically in RTN neurons, the authors' solid results show impaired ventilatory responses to elevated inspired CO2, measured by whole-body plethysmography in freely behaving adult rats, that develop progressively over a four-week period in vivo, indicating effects on RTN neuron transcriptional activity and associated blunting of the CO2 ventilatory response. The RTN neuronal mRNA expression data presented suggests the impaired hypercapnic ventilatory response is possibly due to the decreased expression of key proton sensors in the RTN. This study will be of interest to neuroscientists studying respiratory neurobiology as well as the neurodevelopmental control of motor behavior.

Strengths:

(1) The authors used a shRNA viral approach to progressively knock down the PHOX2B protein, specifically in RTN neurons, to determine whether PHOX2B is necessary for the survival and/or chemosensory function of adult RTN neurons in vivo.

(2) To determine the extent of PHOX2B knockdown in RTN neurons, the authors combined RNAScope and immunohistochemistry assays to quantify the subpopulation of RTN neurons expressing PHOX2B and Neuromedin B (Nmb), which has been proposed to be key chemosensory neurons in the RTN.

(3) The authors demonstrate that knockdown efficiency is time-dependent, with a progressive decrease in the number of Nmb-expressing RTN neurons that co-express PHOX2B over a four-week period.

(4) Their results convincingly show hypoventilation, particularly in 7.2% CO2 only, for PHOX2B-shRNA RTN-injected rats after four weeks compared to naïve and non-PHOX2B-shRNA targeted (NT-shRNA) RTN-injected rats, suggesting a specific impairment of chemosensitive properties in RTN neurons with PHOX2B knockdown.

(5) Analysis of the association between PHOX2B knockdown in RTN neurons and the

attenuation of the hypercapnic ventilatory response (HCVR), by evaluating the correlation between the number of Nmb+/PHOX2B+ or Nmb+/PHOX2B- cells in the RTN and the resulting HCVR, showed a significant correlation between HCVR and number of Nmb+/PHOX2B+ and Nmb+/PHOX2B- cells, suggesting that the number of PHOX2B-expressing cells in the RTN is a predictor of the chemoreflex response and the reduction of PHOX2B protein impairs the CO2-chemoreflex.

(6) The data presented indicate that PHOX2B knockdown reduces the HCVR and the expression of Gpr4 and Task2 mRNAs. This suggests that PHOX2B knockdown affects RTN neurons' transcriptional activity and decreases the CO2 response, possibly by reducing the expression of key proton sensors in the RTN.

(7) This study's results show that independent of its role during development, PHOX2B is still required to maintain proper CO2 chemoreflex responses in the adult brain, and its reduction in CCHS may contribute to the respiratory impairment in this disorder.

Weaknesses:

(1) The authors found a significant decrease in the total number of Nmb+ RTN neurons (i.e., Nmb+/PHOX2B+ plus Nmb+/ PHOX2B-) in NT-shRNA rats at two weeks post viral injection, and also at the four-week period where the impairment of the chemosensory function of the RTN became significant, suggesting some inherent cell death possibly due to off-target toxic effects associated with shRNA procedures.

(2) The tissue sampling procedures for quantifying numbers of cells expressing proteins/mRNAs throughout the extended RTN region bilaterally have not been completely validated to accurately represent the full expression patterns in the RTN under the experimental conditions.

(3) The inferences about RTN neuronal expression of NMB, GPR4, or TASK2 are based on changes in mRNA levels, so it remains speculation that the observed reduction in Gpr4 and Task2 mRNA translates to a reduction in the protein levels and associated reduction of RTN neuronal chemosensitive properties.

---

## [Referee Report · Reviewer #2 (Public Review)]

Summary:

This significant research explored how the PHOX2B transcription factor functions within neurons located in the retrotrapezoid nucleus (RTN), a crucial brainstem chemosensory area, to sustain appropriate CO2 chemoreflex reactions related to breathing in adult rats when observed in a living state. By applying a viral shRNA technique to selectively suppress PHOX2B in RTN neurons, the authors present compelling evidence of deteriorating ventilatory reactions to increased CO2 levels. This impairment progresses over a four-week period in vivo, hinting at disruptions in RTN neuron transcriptional processes and a consequent dulling of CO2-induced breathing responses. The data on RTN neuronal mRNA expression indicates that the weakened hypercapnic ventilatory response may stem from reduced levels of crucial proton sensors within the RTN. This research holds relevance for neuroscientists focused on the neurobiology of respiration and the neurodevelopmental regulation of motor functions.

Strengths:

The authors employed a shRNA viral strategy to systematically reduce PHOX2B protein levels, targeting RTN neurons specifically, to assess the importance of PHOX2B for the survival and chemosensory capabilities of adult RTN neurons in a living organism. The findings of this research underscore that beyond its developmental role, PHOX2B remains essential for sustaining accurate CO2 chemoreflex reactions in the adult brain. Furthermore, its diminished presence in Congenital Central Hypoventilation Syndrome (CCHS) could be a factor in the respiratory deficiencies observed in the condition. This study highlights the critical ongoing function of PHOX2B in adult physiology and its potential impact on respiratory health, offering valuable insights for the scientific and medical communities involved in treating and understanding respiratory disorders.

Weaknesses:

N/A

---

## [Referee Report · Reviewer #3 (Public Review)]

A brain region called the retrotrapezoid nucleus (RTN) regulates breathing in response to changes in CO2/H+, a process termed central chemoreception. A transcription factor called PHOX2B is important for RTN development and mutations in the PHOX2B gene result in a severe type of sleep apnea called Congenital Central Hypoventilation Syndrome. PHOX2B is also expressed throughout life, but its postmitotic functions remain unknown. This study shows that knockdown of PHOX2B in the RTN region in adult rats decreased expression of Task2 and Gpr4 in Nmb-expressing RTN chemoreceptors and this corresponded with a diminished ventilatory response to CO2 but did not impact baseline breathing or the hypoxic ventilatory response. These results provide novel insight regarding postmitotic functions of PHOX2B in RTN neurons.

I have two main concerns and several points of clarification.

Main issues:

(1) The experimental approach was not targeted to Nmb+ neurons and since other cells in the area also express Phox2b, conclusions should be tempered to focus on Phox2b expressing parafacial neurons NOT specifically RTN neurons

(2) It's not clear whether PHOX2B is important for transcription of pH sensing machinery, cell health or both. If knockdown of PHOX2B knockdown results in loss of RTN neurons this is also expected to decrease Task2 and Gpr4 levels, albeit by a transcription-independent mechanism.

Other points:

(3) All individual data points should be visible in floating bar graphs in Figs 1 and 4. For example, I don't see any dots for naïve animals in any of the panels in Fig. 1.

(4) the C1 and facial partly overlap with the RTN at this level of the medulla and these cells should appear as Phox2b+/Nmb- cells so it is not clear to me why these cells are not evident in the control tissue in figs 2B and 3B. Also, some of the bregma levels shown in Fig. 5A overlap with Figs 2-3 so again it's not clear to me how this non-cell type specific viral approach was targeted to Nmb cells but not near by TH+ cells. Please clarify.

(5) How do you get a loss of Nmb+ neurons (Figs 2-3) with no change in Nmb fluorescence (Fig. 5B)? In the absence of representative images these results are not compelling and should be substantiated by more readily quantifiable approaches like qPCR.

---

## [Author Response]

The following is the authors’ response to the original reviews.

**eLife assessment**
This important study utilizes a virus-mediated short hairpin RNA (shRNA) approach to investigate in a novel way the role of the wild-type PHOX2B transcription factor in critical chemosensory neurons in the brainstem retrotrapezoid nucleus (RTN) region for maintaining normal CO2 chemoreflex control of breathing in adult rats. The solid results presented show blunted ventilation during elevated inhaled CO2 (hypercapnia) with knockdown of PHOX2B, accompanied by a reduction in expression of Gpr4 and Task2 mRNA for the proposed RTN neuron proton sensor proteins GPR4 and TASK2. These results suggest that maintained expression of wild-type PHOX2B affects respiratory control in adult animals, which complements previous studies showing that PHOX2B-expressing RTN neurons may be critical for chemosensory control throughout the lifespan and with implications for neurological disorders involving the RTN. When some methodological, data interpretation, and prior literature reference issues further highlighting novelty are adequately addressed, this study will be of interest to neuroscientists studying respiratory neurobiology as well as the neurodevelopmental control of motor behavior.
**Public Reviews:**

**Reviewer #1 (Public Review):**
Summary:This important study investigated the role of the PHOX2B transcription factor in neurons in the key brainstem chemosensory structure, the retrotrapezoid nucleus (RTN), for maintaining proper CO2 chemoreflex responses of breathing in the adult rat in vivo. PHOX2B has an important transcriptional role in neuronal survival and/or function, and mutations of PHOX2B severely impair the development and function of the autonomic nervous system and RTN, resulting in the developmental genetic disease congenital central hypoventilation syndrome (CCHS) in neonates, where the RTN may not form and is functionally impaired. The function of the wild-type PHOX2B protein in adult RTN neurons that continue to express PHOX2B is not fully understood. By utilizing a viral PHOX2B-shRNA approach for knockdown of PHOX2B specifically in RTN neurons, the authors' solid results show impaired ventilatory responses to elevated inspired CO2, measured by whole-body plethysmography in freely behaving adult rats, that develop progressively over a four-week period in vivo, indicating effects on RTN neuron transcriptional activity and associated blunting of the CO2 ventilatory response. The RTN neuronal mRNA expression data presented suggests the impaired hypercapnic ventilatory response is possibly due to the decreased expression of key proton sensors in the RTN. This study will be of interest to neuroscientists studying respiratory neurobiology as well as the neurodevelopmental control of motor behavior.Strengths:(1) The authors used a shRNA viral approach to progressively knock down the PHOX2B protein, specifically in RTN neurons to determine whether PHOX2B is necessary for the survival and/or chemosensory function of adult RTN neurons in vivo.(2) To determine the extent of PHOX2B knockdown in RTN neurons, the authors combined RNAScope and immunohistochemistry assays to quantify the subpopulation of RTN neurons expressing PHOX2B and neuromedin B (Nmb), which has been proposed to be key chemosensory neurons in the RTN.(3) The authors demonstrate that knockdown efficiency is time-dependent, with a progressive decrease in the number of Nmb-expressing RTN neurons that co-express PHOX2B over a four-week period.(4) Their results convincingly show hypoventilation particularly in 7.2% CO2 only for PHOX2B-shRNA RTN-injected rats after four weeks as compared to naïve and non-PHOX2B-shRNA targeted (NT-shRNA) RTN injected rats, suggesting a specific impairment of chemosensitive properties in RTN neurons with PHOX2B knockdown.(5) Analysis of the association between PHOX2B knockdown in RTN neurons and the attenuation of the hypercapnic ventilatory response (HCVR), by evaluating the correlation between the number of Nmb+/PHOX2B+ or Nmb+/PHOX2B- cells in the RTN and the resulting HCVR, showed a significant correlation between HCVR and number of Nmb+/PHOX2B+ and Nmb+/PHOX2B- cells, suggesting that the number of PHOX2B-expressing cells in the RTN is a predictor of the chemoreflex response and the reduction of PHOX2B protein impairs the CO2-chemoreflex.(6) The data presented indicate that PHOX2B knockdown not only causes a reduction in the HCVR but also a reduction in the expression of Gpr4 and Task2 mRNAs, suggesting that PHOX2B knockdown affects RTN neurons transcriptional activity and decreases the CO2 response, possibly by reducing the expression of key proton sensors in the RTN.(7) Results of this study show that independent of the role of PHOX2B during development, PHOX2B is still required to maintain proper CO2 chemoreflex responses in the adult brain, and its reduction in CCHS may contribute to the respiratory impairment in this disorder.Weaknesses:(1) The authors found a significant decrease in the total number of Nmb+ RTN neurons (i.e., Nmb+/PHOX2B+ plus Nmb+/ PHOX2B-) in NT-shRNA rats at two weeks post viral injection, and also at the four-week period where the impairment of the chemosensory function of the RTN became significant, suggesting some inherent cell death possibly due to off-target toxic effects associated with shRNA procedures that may affect the experimental results.(2) The tissue sampling procedures for quantifying numbers of cells expressing proteins/mRNAs throughout the extended RTN region bilaterally have not been completely validated to accurately represent the full expression patterns in the RTN under experimental conditions.(3) The inferences about RTN neuronal expression of NMB, GPR4, or TASK2 are based on changes in mRNA levels, so it remains speculation that the observed reduction in Gpr4 and Task2 mRNA translates to a reduction in the protein levels and associated reduction of RTN neuronal chemosensitive properties.

Thank you for sharing the excitement for our study showing novel findings on the contribution of PHOX2B to the chemoreflex response and activity of adult RTN neurons. We believe that reporting the results on cell death following shRNA viral injections, potentially due to some off-target effects, are important to share with the scientific community to help plan experiments of similar kind in various fields of neuroscience.

Thanks for pointing out your concerns about cell quantification, we have edited the methods and results section to add clarity about our analytical procedure.

As we discussed in the manuscript, we were only able to assess mRNA levels of Nmb, Gpr4, Task2 as current available antibodies for the 3 targets are still unreliable. Future studies will benefit from the analysis of changes at protein levels and possibly electrophysiological recordings to verify that chemosensitive properties of RTN neurons are impaired due to reduction of PHOX2B expression. We discuss these limitations in the discussion.

**Reviewer #2 (Public Review):**
Summary:The authors used a short hairpin RNA technique strategy to elucidate the functional activity of neurons in the retrotrapezoid nucleus (RTN), a critical brainstem region for central chemoreception. Dysfunction in this area is associated with the neuropathology of congenital central hypoventilation syndrome (CCHS). The subsequent examination of these rats aimed to shed light on the intricate aspects of RTN and its implications for central chemoreception and disorders like CCHS in adults. They found that using the short hairpin RNA (shRNA) targeting Phox2b mRNA, a reduction of Phox2b expression was observed in Nmb neurons. In addition, Phox2b knockdown did not affect breathing in room air or under hypoxia, but the hypercapnia ventilatory response was significantly impaired. They concluded that Phox2b in the adult brain has an important role in CO2 chemoreception. They thought that their findings provided new evidence for mechanisms related to CCHS neuropathology. The conclusions of this paper are well supported by data, but careful discussion seems to be required for comparison with the results of various previous studies performed by different genetic strategies for the RTN neurons.Strengths:The most exciting aspect of this work is the modelling of the Phox2b knockdown in one element of the central neuronal circuit mediating respiratory reflexes, that is in the RTN. To date, mutations in the PHOX2B gene are commonly associated with most patients diagnosed with CCHS, a disease characterized by hypoventilation and absence of chemoreflexes, in the neonatal period, which in severe cases can lead to respiratory arrest during sleep. In the present study, the authors demonstrated that the role of Phox2b extends beyond the developmental period, and its reduction in CCHS may contribute to the respiratory impairment observed in this disorder.Weaknesses:Whereas the most exciting part of this work is the knockdown of the Phox2b in the RTN in adult rodents, the weakness of this study is the lack of a clear physiological, developmental, and anatomical distinction between this approach and similar studies already reported elsewhere (Ruffault et al., 2015, DOI: 10.7554/eLife.07051; Ramanantsoa et al., 2011, DOI: 10.1523/JNEUROSCI.1721-11.2011; Huang et al., 2017, DOI: 10.1016/j.neuron.2012.06.027; Hernandez-Miranda et al., 2018, DOI: 10.1073/pnas.1813520115; Ferreira et al., 2022 DOI: 10.7554/eLife.73130; Takakura et al., 2008 DOI: 10.1113/jphysiol.2008.153163; Basting et al., 2015 DOI: 10.1523/JNEUROSCI.2923-14.2015; Marina et al., 2010 DOI: 10.1523/JNEUROSCI.3141-10.2010). In addition, several conclusions presented in this work are not directly supported by the provided data.

Thanks for the feedback on or manuscript. We have further highlighted in our discussion the previous developmental work aimed at determining the role of PHOX2B in embryonic development. Our study was triggered by the fascinating observations that despite its important role in development of the central and peripheral nervous system, PHOX2B is still present in the adult brain and its function in adult neurons is unknown, thus we aimed to investigate its role in the adult RTN by knocking down its expression with a shRNA approach. Therefore, in our model knockdown of PHOX2B does not affect development of the RTN. Previous studies (mentioned by the reviewer, as well as cited in the manuscript) have focused on investigating (1) the role of PHOX2B in the developmental period, (2) the physiological changes associated with the transgenic expression of mutant forms of PHOX2B in relation to CCHS, (3) the killing or the acute silencing/excitation of neuronal activity of PHOX2B+ RTN neurons. Our study had a different aim: to test whether the transcription factor PHOX2B had a physiologically relevant role in adult RTN neurons. In this experimental approach PHOX2B is not altered throughout embryonic or postnatal development. By knocking down PHOX2B in the Nmb+ cells of the RTN our results show a reduction in chemoreflex response and mRNA expression of protein sensors. Hence, we conclude that PHOX2B alters the function of Nmb+ RTN neurons, possibly through transcriptional changes including the reduction in Gpr4 and Task2 mRNA expression.

**Reviewer #3 (Public Review):**
A brain region called the retrotrapezoid nucleus (RTN) regulates breathing in response to changes in CO2/H+, a process termed central chemoreception. A transcription factor called PHOX2B is important for RTN development and mutations in the PHOX2B gene result in a severe type of sleep apnea called Congenital Central Hypoventilation Syndrome. PHOX2B is also expressed throughout life, but its postmitotic functions remain unknown. This study shows that knockdown of PHOX2B in the RTN region in adult rats decreased expression of Task2 and Gpr4 in Nmb-expressing RTN chemoreceptors and this corresponded with a diminished ventilatory response to CO2 but did not impact baseline breathing or the hypoxic ventilatory response. These results provide novel insight regarding the postmitotic functions of PHOX2B in RTN neurons.Main issues:(1) The experimental approach was not targeted to Nmb+ neurons and since other cells in the area also express Phox2b, conclusions should be tempered to focus on Phox2b expressing parafacial neurons NOT specifically RTN neurons.(2) It is not clear whether PHOX2B is important for the transcription of pH sensing machinery, cell health, or both. If knockdown of PHOX2B knockdown results in loss of RTN neurons this is also expected to decrease Task2 and Gpr4 levels, albeit by a transcription-independent mechanism.

Although we did not specifically target Nmb+ neurons, we performed viral injections within the area where neurons expressing PHOX2B and Nmb are localized (i.e., the RTN region). We carefully quantified the impact of PHOX2B knockdown on Nmb expressing neurons, as well as the effects on the adjacent TH expressing C1 population and FN neurons (figure 5). As reported in the results section, significant changes in the numbers of PHOX2B expressing neurons was only observed at the site of injection in PHOX2B+/Nmb+ neurons. We did not observe changes in the total number of C1 cells (TH+/PHOX2B+), in the number of TH cells coexpressing PHOX2B, or in the hypoxic ventilatory response (which is dependent on the health status of C1 neuron). We have updated figure 5 to show representative expression of PHOX2B in TH+ neurons in the ventral medulla to complement our cell count analysis. To address potential effects on other cell populations we have edited our discussion as follows:

“PHOX2B knockdown was also restricted to RTN neurons, as adjacent C1 TH+ neurons did not show any change in number of TH+/PHOX2B+ expressing cells, although we cannot exclude that some C1 cells may have been infected and their relative PHOX2B expression levels were reduced. To support the lack of significant alterations associated with the possible loss of C1 function was the absence of significant changes in the hypoxic response that has been shown to be dependent on C1 neurons (Malheiros-Lima et al., 2017).”

Where appropriate, we have substituted “RTN” with “Nmb expressing neurons of the RTN” throughout the manuscript.

We have clarified in the methods and results section how we quantified Task2 and Gpr4 mRNA expression. The quantification was performed on a pool of single cells (200-250/rat) expressing Nmb. Hence, the overall reduction is not a result of general fluorescence loss in the RTN region, but specifically assessed in single cells expressing Nmb. This is therefore independent of the reduction of the total number of Nmb cells.

We propose that cell death is not a direct effect of PHOX2B knockdown, but rather it is associated with the injection of the viral constructs that have been already reported to promote some off-target effects (as reported in the manuscript). While modest cell death is observed only in the first two weeks post-infection, it does not increase further between 2 and 4 weeks post infection, when the reduction in PHOX2B (not associated with a further reduction in Nmb+ cells, hence no further cell death in RTN) is evident and the respiratory chemoreflex is impaired. These results suggest that (1) reduction of PHOX2B is not responsible for cell death; (2) it is the reduction of PHOX2B levels that promotes chemoreflex impairment. Given the observation that Nmb cells with no detectable PHOX2B protein show reduced expression of Task2 and Gpr4 mRNA, we propose that one of the possible mechanisms of chemoreflex impairment in PHOX2B shRNA rats is the reduction of Task2 and Gpr4. In the discussion we also suggest possible additional mechanisms that can be investigated in further studies.

**Recommendations for the authors:**.In revising this manuscript, the authors should carefully address the issues raised by the reviewers to substantially improve the manuscript and solidify the reviewers' general assessment of the potential importance of this work.
**Reviewer #1 (Recommendations For The Authors):**
Major concerns:(1) The cell counts for Nmb+/PHOX2B+ and Nmb+/PHOX2B- RTN neurons are a critical component of the study, and it is unclear how the tissue sampling procedures (eight sections per animal) for quantifying numbers of cells expressing proteins/mRNAs throughout the extended RTN region bilaterally has been validated to accurately represent the full expression patterns in the RTN under the experimental conditions. It is possible that the sampling/quantification procedures used may be adequate, but validation is important. Also, quantification of the CTCF signal for Nmb, Gpr4, and Task2 mRNA is an important component of this study, but only four sections/rats were used.

Thank you for pointing out your concern on our quantification method. We have clarified in the methods section the procedure for cell counting and quantification of the CTCF signal. We have sampled the area of the RTN in order to identify Nmb cells of RTN.

We have edited the methods section as follows:

“To quantify Nmb+/PHOX2B- and Nmb+/PHOX2B+ neurons within the RTN region, we analysed one in every seven sections (210 µm interval; 8 sections/rat in total) along the rostrocaudal distribution of the RTN on the ventral surface of the brainstem and compared total bilateral cell counts of PHOX2B-shRNA rats with non-target control (NT-shRNA) and naïve rats. Cells that expressed Nmb and Phox2b mRNAs but did not show co-localization with PHOX2B protein were considered Nmb+/PHOX2B-.

The Corrected Total Cell Fluorescence (CTCF) signal for Nmb, Gpr4 and Task2 mRNAs was quantified as previously described (Cardani et al., 2022; McCloy et al., 2014). Briefly, a Leica TCS SP5 (B-120G) Laser Scanning Confocal microscope was used to acquire images of the tissue. Exposure time and acquisition parameters were set for the naïve group and kept unchanged for the entire dataset acquisition. The collected images were then analysed by selecting a single cell at a time and measuring the area, integrated density and mean grey value (McCloy et al., 2014). For each image, three background areas were used to normalize against autofluorescence. We used 4 sections/rat (210 µm interval) to count Nmb, Gpr4 and Task2 mRNA CTCF in the core of the RTN area where several Nmb cells could be identified. For each section two images were acquired with a 20× objective, so that at least fifty cells per tissue sample were obtained for the mRNA quantification analysis. To evaluate changes in Nmb mRNA expression levels following PHOX2B knockdown at the level of the RTN, we compared, the fluorescence intensity of each RTN Nmb+ cell (223.2 ± 37.1 cells/animal) with the average fluorescent signal of Nmb+ cells located dorsally in the NTS (4.3 ± 1.2 cells/animal) (Nmb CTCF ratio RTN/NTS) as we reasoned that the latter would not be affected by the shRNA infection and knockdown.

To quantify Gpr4 and Task2 mRNA expression in Nmb cells of the RTN, we first quantified single cell CTCF for either Gpr4 (200.7 ± 13.2 cells/animal) or Task2 (169.6 ± 10.3 cells/animal) mRNA in Nmb+ RTN neurons in the 3 experimental groups (naïve, NT shRNA and PHOX2B shRNA) independent of their PHOX2B expression. We then compared CTCF values of Gpr4 and Task2 mRNA between Nmb+/PHOX2B+ and Nmb+/PHOX2B- RTN neurons in PHOX2B-shRNA rats to address changes in their mRNA expression induced by PHOX2B knockdown.

(2) Furthermore, to evaluate changes in Nmb mRNA expression following PHOX2B knockdown at the level of the RTN, it is stated in Materials and Methods "we compared, on the same tissue section, the fluorescence intensity of RTN Nmb+ cells with the signal of Nmb+ cells in the NTS (Nmb CTCF ratio RTN/NTS)". How this was accomplished is unclear, considering the non-overlapping locations of the RTN and rostral NTS. Providing images would be helpful.

The first sections containing Nmb cells in the ventral medulla also express few Nmb cells in the dorsal medulla. We used those cells as reference for fluorescence levels since they would not be affected by the viral infection. Similar cells are also present in the brains of mice and reported in the Allen Brain atlas (https://mouse.brain-map.org/experiment/show/71836874). We have clarified our procedure in the methods section (see above) and included a sample image of Nmb in both ventral and dorsal regions in Figure 5.

(3) The staining for tyrosine hydroxylase (TH) to identify and quantify C1 cells (TH+/PHOX2B+) following shRNA injection provides important information, and it would be useful to show images of histological examples to accompany Fig. 5A.

We included in figure 5A a sample image of C1 neurons used for our TH quantification.

Minor:(1) Provide animal ns in the text of the Results section for the four weeks of PHOX2B knockdown.

They have been included.

(2) Please state in the legends for Figures 2 & 3, which images are superimposition images.

We have in the figure information about merged images.

**Reviewer #2 (Recommendations For The Authors):**
This manuscript by Cardani and colleagues attempts to address whether a reduction in Phox2b expression in chemosensitive neuromedin-B (NMB)-expressing neurons in the RTN alters respiratory function. The authors used a short hairpin RNA technique to silence RTN chemosensor neurons. The present study is very interesting, but there are several major concerns that need to be addressed, including the main hypothesis.Major(1) Page 6, lines 119-121: I did not grasp the mechanistic property described by the authors in this passage, nor did I understand the experiments they conducted to establish a mechanistic link between Phox2b and the chemosensitive property. Could the authors provide further clarification on these points?

We believe the reviewer refers to this paragraph: “In order to have a better understanding of the role of PHOX2B in the CO2 homeostatic processes we used a non-replicating lentivirus vector of two short-hairpin RNA (shRNA) clones targeting selectively Phox2b mRNA to knockdown the expression of PHOX2B in the RTN of adult rats and tested ventilation and chemoreflex responses. In parallel, we also determined whether knockdown of PHOX2B in adult RTN neurons negatively affected cell survival. Finally, we sought to provide a mechanistic link between PHOX2B expression and the chemosensitive properties of RTN neurons, which have been attributed to two proton sensors, the proton-activated G protein-coupled receptor (GPR4) and the proton-modulated potassium channel (TASK-2).”

The rationale for running these experiments is based on the fact that it is well known in the literature that PHOX2B is an important transcription factor for the development of several neuronal populations. PHOX2B Knockout mice die before birth and heterozygous mice have some anatomical defects, but respiration is only impaired in the early post-natal period. While many developmental transcription factors are generally downregulated in the post-natal period, PHOX2B is still expressed in some neurons into adulthood. What is the function of PHOX2B in these fully developed neurons? We do not know as we do not yet know the entire set of target genes that PHOX2B regulates in the adult brain. Hence we decided to test what would happen if we knocked down the PHOX2B protein in the Nmb neurons of the RTN, an area that is critical for central chemoreception and involved in the presentation of CCHS. Our results show that reduction of PHOX2B blunts the CO2 chemoreflex response and reduces mRNA expression of Task2 and Gpr4, two pH sensors that have been shown to be key for RTN chemosensitive properties. We also show that the Nmb mRNA and cell survival are not affected by PHOX2B knockdown and we propose that the reduced CO2 chemoreflex may be attributed to a reduction of chemosensory function of Nmb neurons of the RTN due to partial loss of Gpr4 and Task2.

(2) It is imperative for the authors to enhance the description of their hypothesis, as, from my perspective, the contribution of the data to the field is not clearly articulated. Numerous more selectively designed experiments were conducted to investigate the role of Phoxb-expressing neurons at the RTN level and their involvement in respiratory activity. In summary, the current study appears to lack novelty.

We respectfully disagree with this statement. We believe we have adequately summarized previous work, although we realize we can’t reference every single publication on this topic. As described above, the developmental role of PHOX2B has been elegantly investigated in mouse embryonic studies (extensively cited in the manuscript). Furthermore, very interesting studies have shown that when the CCHS defining mutant PHOX2B protein (+7Ala PHOX2B) and other mutations linked to CCHS have been transgenically expressed in mice through development, severe anatomical defects are observed and respiratory function is impaired (extensively cited in the manuscript). We have also cited papers relevant to this study that describe the role of PHOX2B/Nmb RTN neurons and the pH protein sensors in the CO2 chemoreflex. If we missed some papers that the reviewer deems essential in the context of this study we will be happy to include them.

We are not aware of other studies that have investigated the specific role of the PHOX2B protein in the adult RTN in the absence of confounding developmental pathogenesis (i.e. in an otherwise ‘healthy’ animal), and of no other studies that looked at the effects on the RTN proton sensors and Nmb expression following PHOX2B knockdown. Hence we believe that our results are novel and, in our opinion, very interesting.

(3) On pages 13 and 14 (Results section), I am seeking clarity on the novelty of the findings. Doug Bayliss's prior work has already demonstrated the role of Gpr4 and Task2 on Phox2b neurons in regulating ventilation in conscious rodents.

Bayliss’ group has elegantly demonstrated that Gpr4 and Task2 are the two proton sensors in the PHOX2B/Nmb neurons of the RTN that have a key role in chemoreception (cited in the manuscript). The novelty of our findings is that we show that a reduction in PHOX2B protein is associated with a reduction of mRNA levels of Gpr4 and Task2. This is a novel finding. Currently, we do not know what transcriptional activity PHOX2B has in adult RTN neurons (i.e., what gene targets PHOX2B has in this cell population and many others) and here we propose that Nmb is not a gene target of PHOX2B while Gpr4 and Task2 are.

(4) The authors assert that the transcription factor Phox2b remains not fully understood. While I concur, the present study falls short of fully investigating the actual contribution of Phox2b to breathing regulation. In other words, the knockdown of Phox2b neurons did not add much to the knowledge of the field.

We respectfully disagree with the reviewer. With the exception of very few target genes, the transcriptional role of PHOX2B beyond the embryonic development is poorly understood. No mechanistic connection has been made before between the transcriptional activity of PHOX2B with the expression of proton sensors in the RTN. Other groups have investigated the role of stimulating or depressing the neuronal activity of PHOX2B/NMB neurons in the RTN showing a key role of RTN on respiratory control, but these prior studies did not test whether changing the expression of the PHOX2B protein in these neurons had a role on respiratory control and the central chemoreflex. No other study has investigated the role of the PHOX2B protein within the RTN cells, with the exception of PHOX2B knockout mice or transgenic expression of the mutated PHOX2B that are relevant for CCHS. Again, these previous studies were done on a background of developmental impairment and to the best of our knowledge did not seek to show any association between PHOX2B expression and expression of Gpr4 or Task2.

(5) I recommend removing the entire section entitled "The role of Phox2b in development and in the adult brain." The authors merely describe Phox2b expression without contextualizing it within the obtained data.

Because reviewers raised the issue about not including important information about the role of PHOX2B in development and respiratory control we prefer to keep the section.

(6) Are the authors aware of whether the shRNA in Phox2b/Nmb neurons truly induced cell death or solely depleted the expression of the transcription factor protein? Do the chemosensitive neurons persist?

This is an excellent question that we tried to address with our study. As we report in figures 2 and 3, we propose that some cell death is occurring as an off-target effect within the first 2 weeks post-infection, likely due to off-target action of the shRNA approach and not dependent on the reduction of PHOX2B expression (discussed in the manuscript). This is further evidenced by our Fig.S1 data in which higher concentrations of shRNA led to more cell death, indicative of off-target effects. We do not believe it is a consequence of our surgical procedure as we do not see similar cell loss when injecting vehicle or other control solutions (unpublished work; Janes et al., 2024).

During the first 2 weeks post-surgery the proportion of Nmb+/PHOX2B- cells does not change compared to control rats or non-target shRNA (knockdown is not yet visible at protein level). Four weeks post-injection, there is no further cell death (assessed by the total number of NMB cells), whereas the fraction of NMB cells that express PHOX2B is reduced (and the fraction of NMB not expressing PHOX2B is increased), suggesting that the reduction of PHOX2B protein in Nmb cells is not correlated with cell loss/survival whereas the impairment that we observe in terms of central chemoreception is possibly due to the progressive decrease of PHOX2B expression in these neurons.

(7) In Figures 2 and 3, it is noteworthy that the authors observe peak expression at a very caudal level. In rats, the RTN initiates at the caudal end of the facial, approximately 11.6 mm, and should exhibit a rostral direction of about 2 mm.

In our experience the Nmb cells on the ventral surface of the medulla peak in number around the caudal tip of the facial nucleus in adult SD rats (Janes et al., 2024). To add clarity to the figure we reported cell count distribution data in relation to the distance from caudal tip of the facial.

Minor(1) I would like to suggest that the authors correct the recurring statement throughout the manuscript that Phox2b is essential only for the development of the autonomic nervous system. In my view, it also plays a crucial role in certain sensory and respiratory systems.

We have addressed this in the manuscript.

(2) Page 4, lines 59-60: Out of curiosity, do the data include information from different countries?

This data refers to information from France and Japan. Currently it is estimated that there are 1000-2000 CCHS patients worldwide.

(3) Page 7, lines 129-131: In my understanding, the sentence is quite clear; if we knock down the PHOX2B gene, we are expected to reduce or even eliminate the expression of Gpr4 or Task2. Am I right?

This is what we propose from the results of this study. We would like to point out that the transcriptional activity of PHOX2B (i.e., what genes PHOX2B regulate) in adult neurons has not yet been fully investigated. With the exception of few target genes (e.g., TH, DBH) the transcriptional activity of PHOX2B in neurons is not yet known. Here we report novel findings that suggest that Gpr4 and Task2 are potential target genes of PHOX2B in RTN neurons.

(4) The authors mentioned that NT-shRNA also impacts CO2 chemosensitivity. Could this effect be attributed to mechanical damage of the tissue resulting from the injection?

Just to clarify, we observe some impairment in chemosensitivity when NT-shRNA was injected in “larger” (2x 200ul/side) volume. No impairment was observed in NT-shRNA when we injected smaller volumes (2x 100ul/side). Physical damage could be a possibility although in our experience (unpublished work; Janes et al, 2024, Acta Physiologica) injections of similar volume of solution performed by the same investigator in the same brain area and experimental settings did not produce a physical lesion associated with respiratory impairment. Hence we attribute the unexpected results with larger volumes to toxic effects associated with the shRNA viral constructs.

(5) In the reference section, the authors should review and correct some entries. For instance, Janes, T. A., Cardani, S., Saini, J. K., & Pagliardini, S. (2024). Title: "Etonogestrel Promotes Respiratory Recovery in an In Vivo Rat Model of Central Chemoreflex Impairment." Running title: "Chemoreflex Recovery by Etonogestrel." Some references contain the journal, pages, and volume, while others lack this information entirely.

We have updated references. Janes et al., 2024 has now been published in Acta Physiologica.

(6) Why does the baseline have distribution points, whereas the other boxplots do not?

We have clarified in the figure legend that, to be fair to the presentation of our results, the data points shown in some of the boxplot graphs do not refer to entire baseline data but only the ones that are outliers.

In our Box-and Whisker-Plots, whiskers represent the 10th and 90th percentiles, showing the range of values for the middle 80% of the data. Individual data values that fall outside the 10th/90th percentile range are represented as single point (outliers).

**Reviewer #3 (Recommendations For The Authors):**
What is the rationale behind dedicating the first paragraph of results to discussing an artifact?

We think that it is important to report off target effects of shRNA viral constructs as concentration and volumes of viruses injected in various studies vary considerably and other investigators may attempt to use larger volumes of viruses to obtain more considerable or faster knockdown but would obtain erroneous conclusions if appropriate tests are not performed.

Furthermore, because some readers could question whether we injected enough virus to knockdown the expression of PHOX2B, and may wonder if with a larger amount of virus we would increase knockdown efficiency, we wanted to show that, in our opinion, we used the maximum amount of virus to knockdown PHOX2B without causing toxic effects or physiological changes that are not dependent on PHOX2B knockdown.

All individual data points should be visible in floating bar graphs in Figures 1 and 4. For example, I don't see any dots for naïve animals in any of the panels in Figure 1.

We have clarified in the figure legend that, to be fair to the presentation of our results, the data points shown in some of the boxplot graphs do not refer to entire baseline data but only the ones that are outliers.

In our Box-and Whisker-Plots, whiskers represent the 10th and 90th percentiles, showing the range of values for the middle 80% of the data. Individual data values that fall outside the 10th/90th percentile range are represented as single point (outliers).

Please include specific F and T values along with DF.

We have included a table with all the specific values in the supplementary section as Table 1.

The C1 and facial partly overlap with the RTN at this level of the medulla and these cells should appear as Phox2b+/Nmb- cells so it is not clear to me why these cells are not evident in the control tissue in Figures 2B and 3B. Also, some of the bregma levels shown in Figure 5A overlap with Figures 2-3 so again it is not clear to me how this non-cell type specific viral approach was targeted to Nmb cells but not nearby TH+ cells. Please clarify.

In our experience, C1 TH cells are located slightly medial to the Nmb cells and they spread much more caudally than Nmb cells of the RTN. We focused our small volume injection in the core of the RTN to target Nmb cells but we also assessed PHOX2B knockdown in TH C1 cells by counting the PHOX2B/TH cells across treatment groups. Although we can’t exclude subtle changes in the C1 population, we did not observe changes in the total number of C1 cells (TH+/PHOX2B+), in the number of TH cells expressing PHOX2B, or in the hypoxic ventilatory response (which is dependent on the health status of C1 neuron). We have updated figure 5 to show representative expression of PHOX2B in TH+ neurons in the ventral medulla to complement our cell count analysis. To address potential effects on other cell populations we have edited our discussion as follows:

“PHOX2B knockdown was also restricted to RTN neurons, as adjacent C1 TH+ neurons did not show any change in number of TH+/PHOX2B+ expressing cells, although we cannot exclude that some C1 cells may have been infected and their relative PHOX2B expression levels were reduced. To support the lack of significant alterations associated with the possible loss of C1 function was the absence of significant changes in the hypoxic response that has been shown to be dependent on C1 neurons (Malheiros-Lima et al., 2017).”

To confirm, Nmb is not expressed in the NTS, and this region was chosen as a background, right?

In order to systematically analyze Nmb mRNA expression we decided to use measurement of fluorescence relative to Nmb neurons present in the dorsal brainstem. Here cells are sparse but we used them as reference fluorescence since they would not be affected by the ventral shRNA injection. Similar cells are also present in the brains of mice and reported by the Allen Brain atlas (https://mouse.brain-map.org/experiment/show/71836874). We have clarified our procedure in the methods section (see above) and included a sample image of Nmb in both ventral and dorsal in Figure 5.

How do you get a loss of Nmb+ neurons (Figs 2-3) with no change in Nmb fluorescence (Fig. 5B)? In the absence of representative images these results are not compelling and should be substantiated by more readily quantifiable approaches like qPCR.

We have clarified in the methods and results section our analytical procedure to assess PHOX2B and Nmb expression. Figure 2 and 3 display the results of counting numbers of Nmb+ cells in the RTN. Figure 5B reports the average of total cell fluorescence measured inside Nmb+ cells, not an average fluorescence measurement of the area of the ventral medulla. Basically, our results show that we have less Nmb cells that express PHOX2B but the overall Nmb mRNA fluorescence (expression) in Nmb cells relative to Nmb fluorescence in cells of the dorsal brainstem is the same.

We have edited the methods as follows:

“The Corrected Total Cell Fluorescence (CTCF) signal for Nmb, Gpr4 and Task2 mRNAs was quantified as previously described (Cardani et al., 2022; McCloy et al., 2014). Briefly, a Leica TCS SP5 (B-120G) Laser Scanning Confocal microscope was used to acquire images of the tissue. Exposure time and acquisition parameters were set for the naïve group and kept unchanged for the entire dataset acquisition. The collected images were then analysed by selecting a single cell at a time and measuring the area, integrated density and mean grey value (McCloy et al., 2014). For each image, three background areas were used to normalize against autofluorescence. We used 4 sections/rat (210 µm interval) to count Nmb, Gpr4 and Task2 mRNA CTCF in the core of the RTN area where several Nmb cells could be identified. For each section two images were acquired with a 20× objective, so that at least fifty cells per tissue sample were obtained for the mRNA quantification analysis. To evaluate changes in Nmb mRNA expression levels following PHOX2B knockdown at the level of the RTN, we compared the fluorescence intensity of each RTN Nmb+ cell (223.2 ± 37.1 cells/animal) with the average fluorescent signal of Nmb+ cells located dorsally in the NTS ( 4.3 ± 1.2 cells/animal) (Nmb CTCF ratio RTN/NTS) as we reasoned that the latter would not be affected by the shRNA infection and knockdown. “

A single cell qPCR analysis would be definitely ideal but a qPCR from dissected tissue would not help us determine whether within a cell there was a reduction in Nmb mRNA levels.

The boxed RTN region in these examples is all over the place. It the RTN should be consistently placed along the ventral surface under the facial and pprox.. equal distance from the trigeminal and pyramids.

We have update the figures to consistently present the areas of interest where Nmb cells are located and images are taken.

Fluorescent in situ typically appears as discrete puncta so it is not clear to me why that is not the case here.

Our images are taken at low magnification (20X) where it is difficult to distinguish the single mRNA molecules. However, is it possible to appreciate the differences between the grainy fluorescent signal in the in situ hybridization assay (RNAScope) and the smoother signal of protein detection in the immunofluorescence assay.

Can TUNEL staining be done to confirm loss of Nmb neurons is due to death and not re-localization?

Does the reviewer mean “cell migration” with relocalization? We do not expect that this would occur in our experiments. Although TUNEL in the first week post-infection could be useful to determine cell death in our tissue, we do not expect a cell migration of neurons within the brain as our viral shRNA injections are performed in adult rats when developmental processes are already concluded.